

# New sensitivity of current LHC measurements to vector-like quarks

Andy Buckley[1], Jonathan Butterworth[2], Louie Corpe[2],
Danping Huang[2] and Puwen Sun[1]

**1** School of Physics & Astronomy, University of Glasgow,
University Place, G12 8QQ, Glasgow, UK
**2** Department of Physics & Astronomy, University College London,
Gower St., WC1E 6BT, London, UK

## Abstract

Quark partners with non-chiral couplings appear in several extensions of the Standard Model. They may have non-trivial generational structure to their couplings, and may be produced either in pairs via the strong and EM interactions, or singly via the new couplings of the model. Their decays often produce heavy quarks and gauge bosons, which will contribute to a variety of already-measured "Standard Model" cross-sections at the LHC. We present a study of the sensitivity of such published LHC measurements to vector-like quarks, first comparing to limits already obtained from dedicated searches, and then broadening to some so-far unstudied parameter regions.



## Contents

*This paper is dedicated to the memory of Puwen Sun, 1990-2020.*

# 1   Introduction

Quark partners with vector-like couplings to the weak sector (VLQs) may exist with masses close enough to the electroweak scale that they can help solve the hierarchy problem, and yet they evade many experimental constraints since their mass is not necessarily acquired via the Brout-Englert-Higgs mechanism [1]. VLQs can be motivated by a variety of higher-energy extensions to the Standard Model (SM), and strategies for discovering them have been developed at both the Tevatron [2] and the LHC [3, 4]. Here we use the model-independent framework presented in Ref. [4]. The allowed parameter space is nevertheless quite tightly constrained by theory, and searches at the LHC have set lower limits of 1 TeV to 1.3 TeV on their masses, for various scenarios within this framework [5–9].

Typically, searches at the LHC have assumed the VLQs couple only to the third generation of SM quarks, since this is the scenario least constrained by previous measurements [4]. For example, as discussed by Buchkremer *et al*, the absence of flavour-changing neutral currents (FCNC) implies strong constraints on the VLQ coupling to the SM when more than one generation couples, but these constraints are eased when the coupling is only to the third generation. Such studies also tend to focus on the production of one flavour of VLQ at a time.

In this paper we examine the sensitivity of available particle-level measurements (as opposed to dedicated searches) to VLQs, using the CONTUR framework [10] [1] to inject signal into results of LHC analyses present in the RIVET library [12, 13], and derive constraints using the CL$_s$ technique [14]. We use Herwig [15,16] to inclusively generate all leading-order $2 \rightarrow 2$ processes involving one or more VLQs. Next-to-leading-order predictions are available [17, 18] but are beyond the scope of this study. This inclusive signal generation implies that a wide array of signatures is covered, allowing us to move away from some of the above simplifying assumptions to provide more general limits[2]. For each parameter point, $30,000$ beyond-the-SM (BSM) events are generated for each beam condition, corresponding to a luminosity significantly greater than that of the data over the relevant parameter space.

We first give an overview of the phenomenology of VLQ models, and how they might be expected to be produced and decay at the LHC in Section 2. We then benchmark CONTUR against the LHC searches in Section 3, using $B^{-\frac{1}{3}}$ and $T^{\frac{2}{3}}$ production only and assuming the $X^{\frac{5}{3}}$ and $Y^{-\frac{4}{3}}$ VLQs to be decoupled. In Section 4, we extend beyond this simplest model, and study the sensitivity for all four VLQs active, with degenerate masses, again for the case of coupling to only the third generation of SM quarks. In Section 5 we look at VLQ production as a function

---

[1]Making use, where possible, of the updated treatment of correlated experimental uncertainties described in Ref. [11].

[2]Since the data are known to agree with the SM, we do expect limits rather than a discovery at this stage!

of the overall coupling $\kappa$ and generic VLQ mass $M_Q$, where single production is of particular interest when allowing non-zero coupling to the first and second generations of SM quarks.

A selection of non-standard VLQ and leptoquark signatures in different models was also studied in contribution 5 of Ref. [11], also showing some interesting sensitivity to these models.

## 2 Overview of VLQ phenomenology

Four types of vector-like quark $Q$ are allowed — $B$, $T$, $X$, and $Y$ — which may be arranged in various weak $SU(2)_L$ multiplets. The $B$ and $T$ are present in all theoretically-allowed scenarios [1, 19] and have EM charges $-\frac{1}{3}$ and $\frac{2}{3}$, like the SM $b$ and $t$, while the $X$ and $Y$ (with charges $\frac{5}{3}$ and $-\frac{4}{3}$ respectively) may be elided in some models. All VLQs have quark-like triplet colour charges. In addition to their quark-like QCD and EM couplings via the usual covariant derivative recipe, the VLQs additionally couple to SM quarks $q$ and weak bosons $V \in \{W, Z, H\}$, via new $QqV$ vertices. These interactions are parametrised by overall couplings $\kappa$, $\xi^V$ parameters controlling the relative strengths of the $V$ couplings to each VLQ, and $\zeta_i$ parameters governing the mix of SM quark generations $i$ in each coupling. The relevant parts of the Lagrangian are, following the notation of Ref. [4],

$$
\begin{aligned}
\mathcal{L} = \kappa_B & \left[ \sqrt{\frac{\zeta_i \xi_W^B}{\Gamma_W^0}} \frac{g}{\sqrt{2}} [\bar{B}_{L/R} W_\mu^- \gamma^\mu u_{L/R}^i] + \sqrt{\frac{\zeta_i \xi_Z^B}{\Gamma_Z^0}} \frac{g}{2c_W} [\bar{B}_{L/R} Z_\mu \gamma^\mu d_{L/R}^i] - \sqrt{\frac{\zeta_i \xi_H^B}{\Gamma_H^0}} \frac{M_B}{v} [\bar{B}_{R/L} H d_{L/R}^i] \right] \\
+ \kappa_T & \left[ \sqrt{\frac{\zeta_i \xi_W^T}{\Gamma_W^0}} \frac{g}{\sqrt{2}} [\bar{T}_{L/R} W_\mu^+ \gamma^\mu d_{L/R}^i] + \sqrt{\frac{\zeta_i \xi_Z^T}{\Gamma_Z^0}} \frac{g}{2c_W} [\bar{T}_{L/R} Z_\mu \gamma^\mu u_{L/R}^i] - \sqrt{\frac{\zeta_i \xi_H^T}{\Gamma_H^0}} \frac{M_T}{v} [\bar{T}_{R/L} H u_{L/R}^i] \right] \\
+ \kappa_X & \left[ \sqrt{\frac{\zeta_i}{\Gamma_W^0}} \frac{g}{\sqrt{2}} [\bar{X}_{L/R} W_\mu^+ \gamma^\mu u_{L/R}^i] \right] + \kappa_Y \left[ \sqrt{\frac{\zeta_i}{\Gamma_W^0}} \frac{g}{\sqrt{2}} [\bar{Y}_{L/R} W_\mu^- \gamma^\mu d_{L/R}^i] \right] + \text{h.c.},
\end{aligned}
$$

(1)

where $M_Q$ is the mass of $Q$, $c_W$ is the usual cosine of the weak mixing angle, $v$ is the Higgs-field vacuum expectation value, and $\Gamma_V^0$ are functions of $m_V/M_Q$ only. The $\kappa$, $\xi$ and $\zeta$ are defined for each $Q$ such that $\sum_V \xi^V = 1$ and $\sum_i \zeta_i = 1$, meaning $\zeta_i \xi^V = \text{BR}(Q \to V q_i)$. Significantly, EM charge conservation implies that the $T$ and $B$ VLQs couple to all three SM weak bosons, while the $X$ and $Y$ couple only to $W^\pm$.

From equation (1) it can be seen that the couplings of $B$ and $X$ to $W$ bosons are the same up to the $\xi^V$ factor: any $W$:$Z$:$H$ mixture other than 1:0:0 will result in a smaller $BWq$ coupling than the $XWq$ one. The same argument applies to the $T$ and $Y$. If we ignore the $\Gamma^0$ components, the $B$ and $T$ couplings to $Z$ and $H$ respectively contain additional factors of $1/\sqrt{2}c_W$ and $M_Q/gv = M_Q/\sqrt{2}m_W$. However, the $\Gamma^0$ factors act to ensure that the $\sum_V \xi^V = 1$ relationships are preserved regardless of $m_V$ and $M_Q$.

This combination of couplings means that the VLQs may be pair-produced from SM initial states via the strong and EM interactions, singly produced via the weak interaction, and weakly pair-produced by $t$-channel exchange of an SM weak boson. They only decay via their weak interaction, into the mixtures of SM weak bosons and quarks governed by the $\xi$ and $\zeta$ parameters. Examples of leading-order Feynman diagrams for VLQ production are shown in Figure 1.

In $pp$ collisions at the LHC, the phenomenology of VLQ production depends strongly on which generations of SM quarks they couple to. In the rest of this section, all four VLQs are

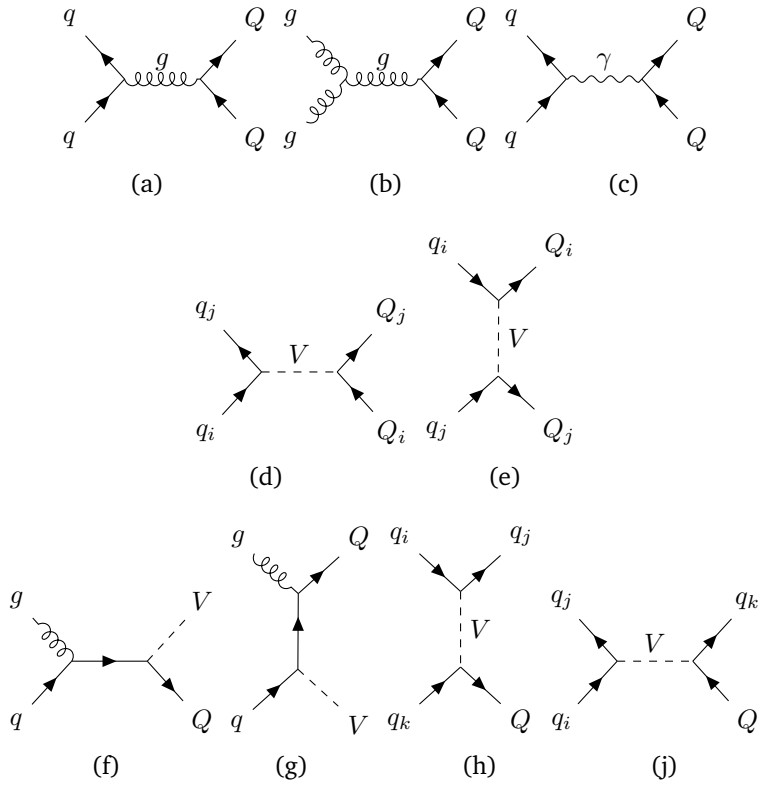

Figure 1: Leading-order Feynman diagrams for production of VLQs $Q$. The top row (a–c) shows VLQ pair-production diagrams via strong and EM interactions, which do not depend on $\kappa$. The second row (d–e) shows pair-production of VLQs via a weak boson $V \in \{W, Z, H\}$, which may lead to different-flavoured VLQs in the final state. The third row (f–i) shows single-production of $Q$ in association with a weak boson or SM quark $q$.

assumed to have the same mass and $\kappa$ values. The $\xi$ parameters are chosen such that the $B$ and $T$ couple to the bosons according to the ratio $W{:}Z{:}H = \frac{1}{2}{:}\frac{1}{4}{:}\frac{1}{4}$, as motivated by Ref. [4].

Pair-production of VLQs typically occurs via SM-like interactions, as shown in Figures 1(a–c). These vertices do not depend on $\kappa$, $\xi$ or $\zeta$, and this feature has been exploited in several LHC analyses to reduce the model-dependence of searches for VLQs [20]. For a VLQ of mass $\sim 1.3$ TeV at the LHC, the cross-section is of the order of 10 fb. It is not, however, strictly accurate that VLQ pair-production is independent of $\kappa$: the diagram in Figure 1e shows that VLQs may be pair-produced in diagrams which involve two $QqV$ vertices, and which therefore have a strong dependence on $\kappa$. Furthermore, it is not even guaranteed that this $QqV$-mediated pair-production be negligible: indeed, it is the only pair-production diagram which can involve two valence quarks in $pp$ collisions. If VLQs couple to first-generation quarks, then Figure 1e can be the dominant production diagram at the LHC, in particular for high $M_Q$, since all others require at least one antiquark or gluon from the proton sea. This effect has already been pointed out, for instance in Ref. [21]. This advantage disappears if the VLQs only couple to second- or third-generation SM quarks. This is illustrated in Figure 2, which shows how the pair-production of $T$ gains a dependence on $\kappa$ if VLQs are allowed to couple to first-generation SM quarks. The effect is analogous for $B$, but differs for $X$ and $Y$ since they can only be produced via Figure 1e if $V$ is a $W$-boson, meaning $XY$ pairs are the only option, and like-flavour valence pairs $uu$ or $dd$ do not contribute. Another interesting feature of the diagrams in Figures 1(d–e)

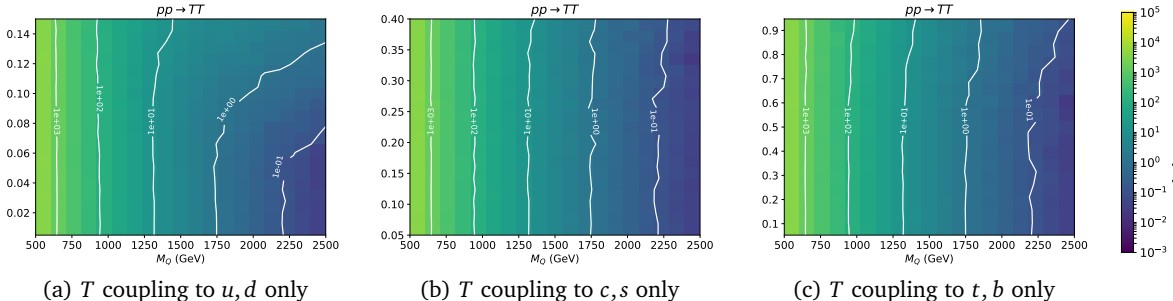

(a) *T* coupling to *u*, *d* only    (b) *T* coupling to *c*, *s* only    (c) *T* coupling to *t*, *b* only

Figure 2: Leading-order cross-sections extracted from Herwig for production of a $TT$ pair as a function of $M_Q$ and $\kappa$, for 13 TeV $pp$ collisions, in the $W{:}Z{:}H = \frac{1}{2}{:}\frac{1}{4}{:}\frac{1}{4}$ scenario, assuming couplings to individual generations of quarks. The white lines indicate the contours for production cross-sections in multiples of 10. The first-generation cross-sections acquire a dependence on $\kappa$ since pair-production initiated by proton valence quarks becomes possible. The situation is analogous for other VLQ flavours, although somewhat attenuated for $X$ and $Y$ since they still require at least one antiquark from the sea to be produced via $W$ exchange in the $t$-channel.

is that the VLQs will be produced with different flavours if $V$ is the $W$-boson, something which is not possible in the other pair-production diagrams. It should be noted that the diagram in Figure 1d contains a model-dependent coupling of a vector boson to pairs of VLQs, the strength of which depends on the electroweak quantum numbers of the multiplet which the VLQs belong to. This diagram is subdominant in the analysis which we perform in this paper, so the particular assumptions we make for this coupling do not affect the overall conclusions.

Single-production of VLQs also has a rich phenomenology. In this case, the production cross-section always has a dependence on $\kappa$ since the $QqV$ vertex must always be involved. Let us consider first the case of VLQ production in association with a weak boson $V$, as shown in Figures 1(f–g). In both diagrams, the process is initiated by a quark and a gluon, and the vector boson is radiated from the quark. The $QqV$ vertex differs depending on the flavour of the VLQ, as described above. The cross-section is also dependent on $\zeta$ however, since the cross-section of Figures 1(f–g) depend strongly on the incoming quark. If the VLQ couples to first-generation quarks, then diagrams where $u$ is incoming will dominate over diagrams with $d$ by a factor of about two. For example, $T + H/Z$ production will be roughly twice as frequent as $B + H/Z$ production, but the situation is reversed for $T + W$, which will be roughly half as frequent as $B + W$ production. The same argument goes for $X + W$ production, which will occur at roughly twice the rate of $Y + W$ production. Overall, in a $W{:}Z{:}H = \frac{1}{2}{:}\frac{1}{4}{:}\frac{1}{4}$ scenario with only first-generation quark couplings $X + V$ would be the dominant process, with $T + V$ or $B + V$ and $Y + V$ occurring $\sim 25\%$ and $\sim 50\%$ less frequently respectively, driven simply by the valence quark populations in the proton. If the VLQs only couple to second-generation quarks, this dependence on the valence quarks disappears, as the diagram can only occur with an incoming $c$ or $s$ from the proton sea. As a result (and leaving aside the effects of available phase space), the production of any flavour of VLQ in association with a $V$ becomes suppressed with respect to pair-production, with the relative cross-sections depending chiefly on the quark PDFs and the $\zeta$ parameters for $B$ and $T$. Finally, for third-generation couplings, $Q + V$ production is further suppressed, and $t$-induced diagrams disappear almost entirely. As a result, $X + V$ production is largely impossible, while $T + V$ may only occur with a $W$. These effects are illustrated in Figure 3, which shows the production cross-sections for $T + V$ and $Y + V$ depending on whether the coupling is to first-, second- or third-generation quarks.

Finally, let us consider production of VLQs in association with quarks. The diagrams for

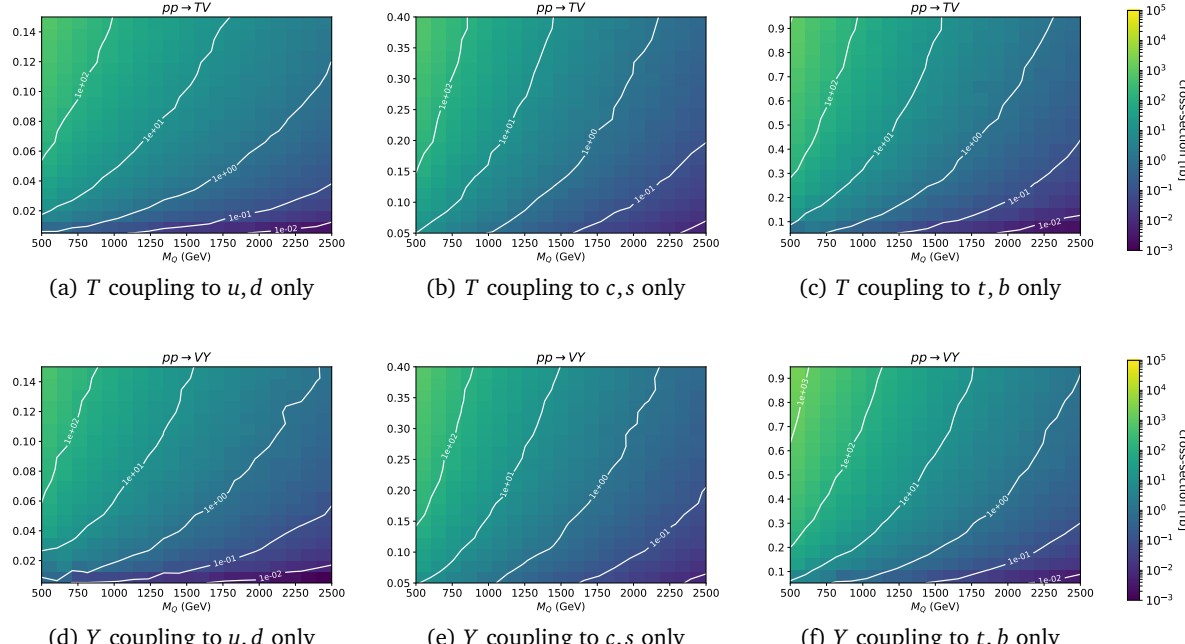

(a) $T$ coupling to $u, d$ only     (b) $T$ coupling to $c, s$ only     (c) $T$ coupling to $t, b$ only

(d) $Y$ coupling to $u, d$ only     (e) $Y$ coupling to $c, s$ only     (f) $Y$ coupling to $t, b$ only

Figure 3: Leading-order cross-sections extracted from Herwig for production of a $T$ and $Y$ with a weak boson as a function of $M_Q$ and $\kappa$, for 13 TeV $pp$ collisions, in the $W{:}Z{:}H = \frac{1}{2}{:}\frac{1}{4}{:}\frac{1}{4}$ scenario, assuming couplings to individual generations of quarks. The white lines indicate the contours for production cross-sections in multiples of 10. First-generation couplings lead to higher production cross-sections, as a result of valence-quark-induced diagrams, while second and third-generation couplings lead to suppressed production rates, according to the relevant quark PDFs.

such processes are shown in Figures 1(h–i), and are mediated by a weak boson, either in the $s$-channel or the $t$-channel. Once again, the importance of this production mechanism at the LHC will depend on the flavours of the incoming quarks, since diagrams involving one or more valence quarks will dominate. The only diagram where both quarks can be valence quarks is the $t$-channel diagram in the scenario where VLQs may couple to first-generation quarks. In this case diagrams involving $uu$ or $ud$ will have the highest cross-sections, while $dd$-induced processes will acquire a suppression factor of about four compared to $uu$ or $ud$, in addition to other considerations such as couplings of the exchanged weak bosons. The dominant process will be $X + q$ production which is induced by $uu$, and which will occur roughly four times more frequently than $Y + q$, which is induced by $dd$. $T + q$ and $B + q$ production cross-sections are also reduced by the fact that the $H$-mediated diagram is suppressed by the very small SM coupling to $u$ or $d$ If the VLQs only couple to second- or third-generation quarks, then the valence-quark-induced processes cease to be available, and $Q + q$ becomes dependent on the quark PDFs, with some production modes becoming almost entirely inaccessible if an initial $t$ quark is involved. This effect is illustrated in Figure 4, where the production cross-sections for $T$ and $X$ in association with a quark are compared.

The overall dominant production processes are unsurprisingly also extremely dependent on the generation(s) which the VLQs are allowed to couple to. If VLQs can couple to first-generation quarks, then by far the dominant process, assuming $W{:}Z{:}H = \frac{1}{2}{:}\frac{1}{4}{:}\frac{1}{4}$, is $X + q$ production, reaching up to $\sim 400$ fb for $M_Q \sim 1$ TeV and $\kappa \sim 0.07$. This is followed by production of other VLQs with quarks, which are about a factor three to four lower in cross-section due to the relative proportion of valence quarks. Gluon-induced or quark-induced VLQ pair-production is typically the next most common process, at $\sim 50$ fb for the same parameter choices, roughly

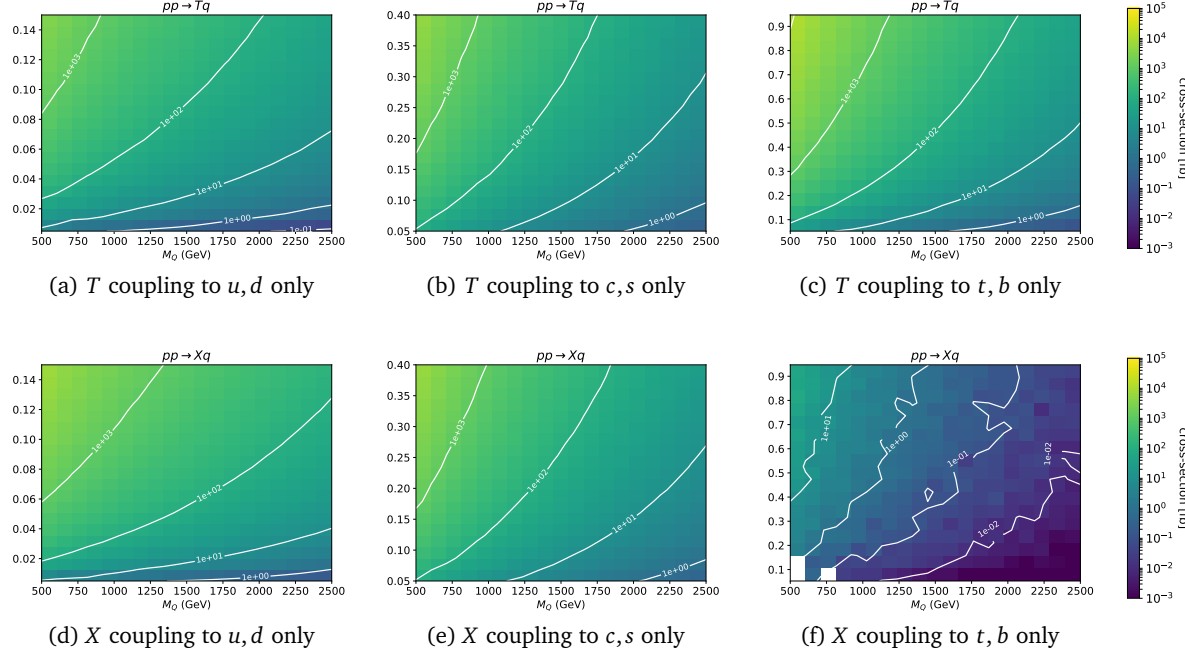

Figure 4: Leading-order cross-sections extracted from Herwig for production of a $T$ and $X$ with a SM quark as a function of $M_Q$ and $\kappa$, for 13 TeV $pp$ collisions, in the $W{:}Z{:}H = \frac{1}{2}{:}\frac{1}{4}{:}\frac{1}{4}$ scenario, assuming couplings to individual generations of quarks. The white lines indicate the contours for production cross-sections in multiples of 10. First-generation couplings lead to higher production cross-sections, as a result of valence-quark-induced diagrams, while second and third-generation couplings lead to suppressed production rates, according to the relevant quark PDFs. $X + q$ production goes from being the dominant production process at the LHC if $X$ couples to first-generation quarks only, to vanishing if $X$ couples to third-generation quarks only. White cells indicate corners of phase-space where the process in question is highly subdominant, and therefore where the cross-section was not sampled during the Herwig run.

twice the rate of VLQ production in association with a weak boson. Since pair-production does not depend too strongly on $\kappa$ at low $M_Q$, there are regions of parameter space, particularly at low $\kappa$, where pair-production may dominate over $Qq$ production. For higher $M_Q$, pair-production from valence quarks involving the $QqV$ vertex may be dominant. If first-generation couplings are forbidden, then pair-production becomes the dominant production process, still at $\sim 50\,\mathrm{fb}$ for the parameter values chosen above. Single-production of VLQs with $V$ or $q$ may still occur, but roughly an order of magnitude less frequently than for first-generation couplings.

The most common experimental signature for VLQ decays is likely to be large numbers of jets: this is unsurprising, given that VLQs decay to quarks and weak bosons (which have their largest branching fractions to quarks). However, such signatures would be swamped by the large QCD background from LHC $pp$ collisions. In many decay chains of the VLQs, $W$-bosons are involved, either directly from the VLQ decay or as a result of the decay of a $t$ quark. This is particularly the case since $X$ and $Y$ VLQs, which are often produced with the highest cross-section, can only decay via $W$. A large fraction of VLQ decays would therefore be expected to produce large missing transverse energy ($E_{\mathrm{T}}^{\mathrm{miss}}$) and one lepton, in addition to multiple jets, some of which may originate from $b$-quarks. In some cases, two leptons may be expected with the missing energy, which can originate from the decay of pair-produced VLQs, with two $W$ bosons decaying leptonically. This signature would exhibit less dependence on $\kappa$

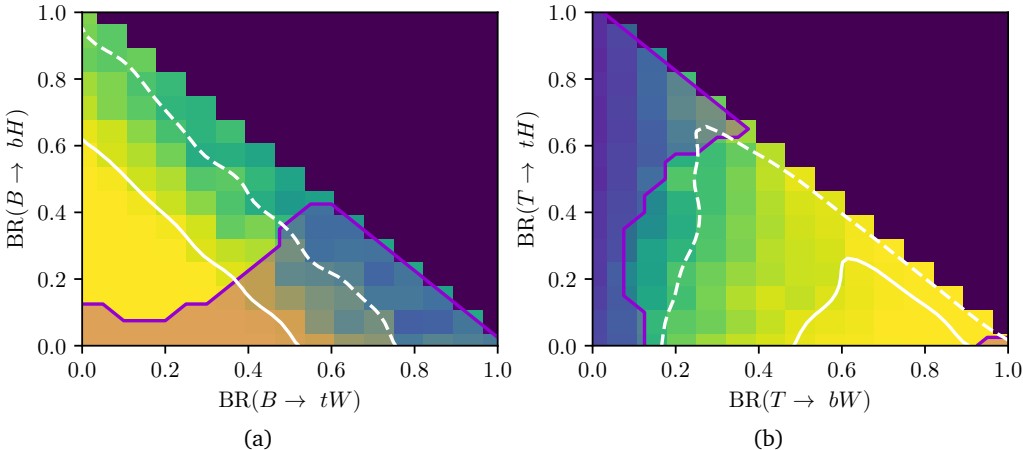

Figure 5: Sensitivity of LHC measurements to (a) $B$-production for $M_B = 1200\,\text{GeV}$ and (b) $T$-production for $M_T = 1350\,\text{GeV}$. The CONTUR exclusion is shown in the bins in which it is evaluated, graduated from yellow through green to black on a linear scale, with the 95% CL (solid white) and 68% CL (dashed white) exclusion contours superimposed. The mauve region is excluded at 95% CL by the ATLAS combination [20].

as a result. Based on this analysis, one can expect LHC analyses targeting $V$+jets or $WW$ final states to be most sensitive to VLQ models.

## 3 Comparison to LHC $B$ and $T$ searches

Assuming all other VLQs decouple from the SM, and that VLQs couple only to the third generation of SM quarks, a $B$ quark with a mass of a few $100\,\text{GeV}$ or more may decay to $Z + b$, $H + b$ or $W + t$, depending upon the relative couplings to these bosons. Similarly a $T$ may decay to $Z + t, H + t$ or $W + b$. Several dedicated searches have been made for these signatures by ATLAS and CMS, giving exclusions up to masses of around a TeV. The ATLAS results in particular are combined and summarised in Ref. [20] and made available in HEPData [22]. CMS results can be found in Ref. [9], with roughly similar sensitivity to the ATLAS results at equivalent parameter points and integrated luminosity. We use the ATLAS results as exemplars for comparison since they are shown for a range of branching fraction values rather than specific points in parameter space. The ATLAS and CMS results each use around $36\,\text{fb}^{-1}$ of $13\,\text{TeV}$ data, while CONTUR uses a range of LHC measurements at $7$, $8$ and $13\,\text{TeV}$ with integrated luminosities between $3$ and $36\,\text{fb}^{-1}$.

Given the subsequent decays of the SM particles produced by the VLQ decays above, BSM events may enter the fiducial phase space of a wide variety of differential cross-sections measured at the LHC, with $b$-jets, $Z, W$+jets, dibosons and multileptons expected to be important, as discussed in the previous section. Many of these measurements are available in RIVET, and are thus accessible to CONTUR. All have been shown to agree with SM calculations, and thus the uncertainties on this agreement provide constraints on the presence of a significant VLQ production cross-section.

Figure 5a shows the CONTUR exclusion region, in the half-plane of different branching ratios, for $M_B = 1200\,\text{GeV}$. This mass places it in middle of the exclusion from ATLAS, which ranges from $1040\,\text{GeV}$ to $1350\,\text{GeV}$ over the half-plane. The mauve region shows the exclusion limit at 95% confidence level (CL) by ATLAS. There are four main searches contributing to

this limit, one each targeting the $Z$ and $H$ decay channels [23, 24], and the remaining two target $B$-decay to $Wt$ [25, 26]. This results in a high sensitivity in the bottom-right corner of the triangle, where $BR(B \to tW)$ is high. The sensitivity in the measurements is somewhat complementary to the searches, and comes primarily from $Z$+jet measurements [27–30].

Figure 5b shows the CONTUR exclusion region for $M_T = 1350$ GeV. The ATLAS search exclusion for this mass value is also shown; in this case, the ATLAS exclusion ranges from 1310 GeV to 1420 GeV over the branching ratio half-plane. Here the difference in sensitivity between the ATLAS searches and CONTUR is nicely seen. From CONTUR, the exclusion comes primarily from measurements involving top quarks and $W$ bosons [31–35]. The ATLAS combination limit in mauve on the other hand, has three contributions from searches sensitive to $T$-decay to $Ht$ [24, 26, 36], two that target $T$-decay to $Zt$ [23, 37], and only one sensitive in the $W$ channel [38]. Again, the measurement sensitivity is quite complementary to the searches.

## 4 Four VLQ flavours

There is no particular reason, other than the desire to take simple benchmarks, why one VLQ should have lower mass than all the others. An equally, and perhaps more, natural scenario is that they all have similar masses. Setting the branching fractions to each gauge boson to be the same for $B$ and $T$ (with $X$ and $Y$ always decaying to $W$ as discussed in Section 2), and assuming still only couplings to third-generation quarks, we have studied the branching fraction half plane for various values of a generic VLQ mass $M_Q$. As might be expected, for a given mass the sensitivity of the measurements is greater than for a single VLQ, with the entire half-plane in the branching fraction space being disfavoured up to about 1800 GeV. Figure 6 shows the results of a scan of the branching fraction of $B$ or $T$ to $qW$ versus VLQ mass $M_Q$, for various multiplet assumptions, and assuming $\mathrm{BF}(Q \to Hq) = \mathrm{BF}(Q \to Zq)$. The difference in exclusion shape between Figures 6a and 6b can be explained by the lack of top density in the proton PDF. The lower parts of the plots only allow single-VLQ production via a neutral boson: for the $T$ this means a $t$-quark would be needed, which is vanishingly rare, and instead QCD/EM pair-production is the only viable mechanism, while the $B$ can be singly-produced via an incoming $b$-quark in the proton sea. In the top half of the plots, the situation is reversed, with $W$-bosons mediating single production, and correspondingly, it is single-$B$ production which is suppressed by the lack of $t$-quarks in the proton sea.

It should be noted that the bounds from direct searches would be enhanced in the presence of multiple degenerate VLQs, so the results shown in Figure 6 should not be compared directly to search results without appropriate re-interpretation.

## 5 All quark generations, and coupling strength

The results discussed in the previous sections apply to scenarios where the VLQs only couple directly to third generation SM quarks and heavy bosons. We now consider cases where the VLQs can couple to other generations of quarks. All production processes described in Section 2 and Figure 1 are included. As discussed in Section 2, at low VLQ masses, the dominant production process is expected to be pair production via the strong interaction, which does not depend on the VLQ–quark coupling, $\kappa$. Despite small values of this coupling suppressing the single-production channels, production of a single VLQ is less kinematically penalised than pair-production and so single-production can dominate at high masses. This is particularly of interest for couplings to first-generation quarks, where amplitudes involving VLQ couplings direct to initial-state quarks become significant due to large high-$x$ valence-quark densities.

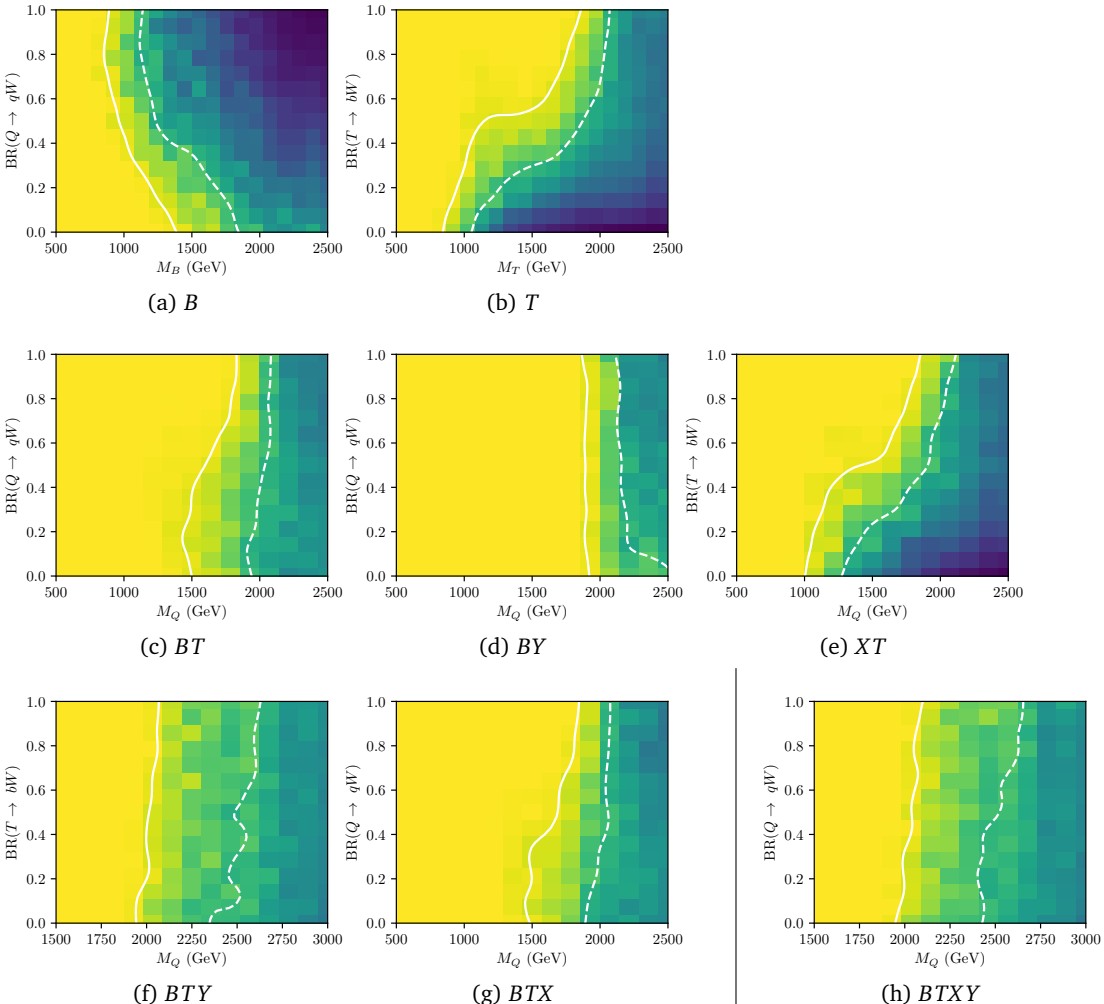

Figure 6: Sensitivity of LHC measurements to VLQ production when all VLQs are degenerate in mass, assuming $\mathrm{BF}(Q \rightarrow Hq) = \mathrm{BF}(Q \rightarrow Zq)$. The CONTUR exclusion is shown in the bins in which it is evaluated, graduated from yellow through green to black on a linear scale, with the 95% CL (solid white) and 68% CL (dashed white) exclusion contours superimposed. The results for various multiplets are shown, in addition to the four-VLQ case which is unrealistic but still a useful benchmark. In the axis labels, $Q$ may refer to $B$ or $T$, while $q$ may refer to $b$ or $t$.

The $t$-channel weak VLQ-pair production process of Figure 1e is a double-recipient of this PDF enhancement for first-generation couplings. On the other hand, the existing, non-collider limits on the coupling between the VLQ and SM quarks are more stringent for the first- and second-generations than for the third. In this section we study the relative impacts of these considerations on constraints from LHC measurements. For the single-light-VLQ representative scenarios considered in Ref. [4], the existing limits on the couplings are estimated as being $\kappa \leq 0.07$ for coupling to the first generation only, $\kappa \leq 0.2$ for the second only, and an order of magnitude smaller when more than one generation is coupled. (For third generation only, $\kappa = 1$ is allowed, as used in the previous section.) These values are indicated in the studies below, although we note that it has been pointed out in Ref. [2] that additional symmetries in specific models may lead to natural cancellations which mean these limits do not apply.

## 5.1 First quark generation

In Figure 7, we show the current LHC measurement sensitivity in the plane of the coupling $\kappa$ to the first generation SM quarks and the VLQ mass $M_Q$, overlaid on a map of the experimental analyses dominating each $(\kappa, M_Q)$ point's $CL_s$ value. These maps are shown for the three extreme VLQ–boson branching-fraction configurations ($W$:$Z$:$H = 0$:$0$:$1, 0$:$1$:$0, 1$:$0$:$0$), and an example admixture of all three bosons ($W$:$Z$:$H = \frac{1}{2}$:$\frac{1}{4}$:$\frac{1}{4}$). In the following sections we will use this same set of $\xi$ configurations to exemplify the LHC $\kappa$–$M_Q$ sensitivities for couplings to the second and third quark generations. The detailed $CL_s$ maps in $\kappa$–$M_Q$, from which these limit contours are constructed, are presented in Appendix B for couplings to each quark generation.

First, we note from the white contour lines that the majority of the first-generation $\kappa$–$M_Q$ plane is excluded at 95% CL, meaning that — despite the lack of dedicated searches — LHC measurements set stringent limits on first-generation VLQs. The tightest limits are set for the $Z$-only $\xi$ configuration in Figure 7b, and the least stringent for the Higgs-only in Figure 7a. At low VLQ masses, below 1 TeV for VLQ decay via a Higgs and below $\sim 1.3$ TeV for decay via a $Z$, QCD and EM pair-production dominates and the exclusion is insensitive to $\kappa$. Above this threshold, a strong $\kappa$ dependence enters via weak $Qq$ production, with the allowed regions of all $W$:$Z$:$H$ configurations extending to the same maximum $\kappa \sim 0.07$ at high mass. This value is similar to the non-collider limits conservatively estimated by Ref. [4] for a mass scale of the order of a TeV from atomic parity violation measurements [39, 40].

The sets of independent measurement analyses which provide the dominant contributions to each $\kappa$-$M_Q$ point in the scans are identified by the background colouring in the plot. This is highly sensitive to the $\xi$ configuration governing VLQ–boson couplings, with the most striking feature being the difference between the $Z$ corner of the $\xi$ triangle (shown in Figure 7b) as compared to the $W$ and $H$ corners (Figures 7c and 7a respectively). The former is unsurprisingly dominated by dilepton (+ jet) analyses, which also make a significant appearance in the excluded $\kappa$-independent region and the un-excluded high-mass region of the $\frac{1}{2}$:$\frac{1}{4}$:$\frac{1}{4}$ mixture (Figure 7d).

Exclusions in the low-mass regions of the $H$ and $W$ corners of the $\xi^V$ triangle (Figures 7a and 7c) are dominated by $WW$ measurements (either direct or via Higgs decays). Interestingly, the dominant contribution to the exclusion power from the $WW$ analysis pool is the measurement of detector-corrected distributions in the *control regions* of the 13 TeV ATLAS leptoquark search [30], an addition to that search study made to enable testing of MC generator models as well as BSM re-interpretation studies. This analysis has exclusion power across the whole mass range, but in the $M_Q = 1$ TeV to 2 TeV region it becomes subdominant to lepton+$E_T^{miss}$ (+ jet) analyses sensitive to single-VLQ production, which drive the exclusion contour downward to enclose lower $\kappa$ values in that mass range. These analyses also dominate the 1 TeV to 2 TeV model exclusions in the mixed $W$:$Z$:$H = \frac{1}{2}$:$\frac{1}{4}$:$\frac{1}{4}$ configuration, and make an appearance at $\kappa \sim 0.05$ to $0.10$ in the $\xi^Z = 1$ configuration. Further investigation of this intrusion of lepton+$E_T^{miss}$+jet analyses into the sea of $WW$-based sensitivity reveals interesting phenomenology: it is dominated by measurements of leading jet $p_T$ in the detector-corrected control regions of the 8 TeV ATLAS vector-boson fusion (VBF) $Wjj$ analysis [31].

Figure 8 shows the $Wjj$ leading-jet $p_T$ distribution for three $W$:$Z$:$H = 1$:$0$:$0$ points near the 95% exclusion contour respectively below, in, and above the single-VLQ-dominated exclusion region. The highest bins of the $p_T$ distribution are seen to be significantly enhanced for non-$Z$-coupling VLQs with masses in the 1 TeV to 2 TeV range, leading to the increased exclusion sensitivity in that region, but for higher masses this excess subsides, due to both the falling cross-section and the excess localising to out-of-range bins.

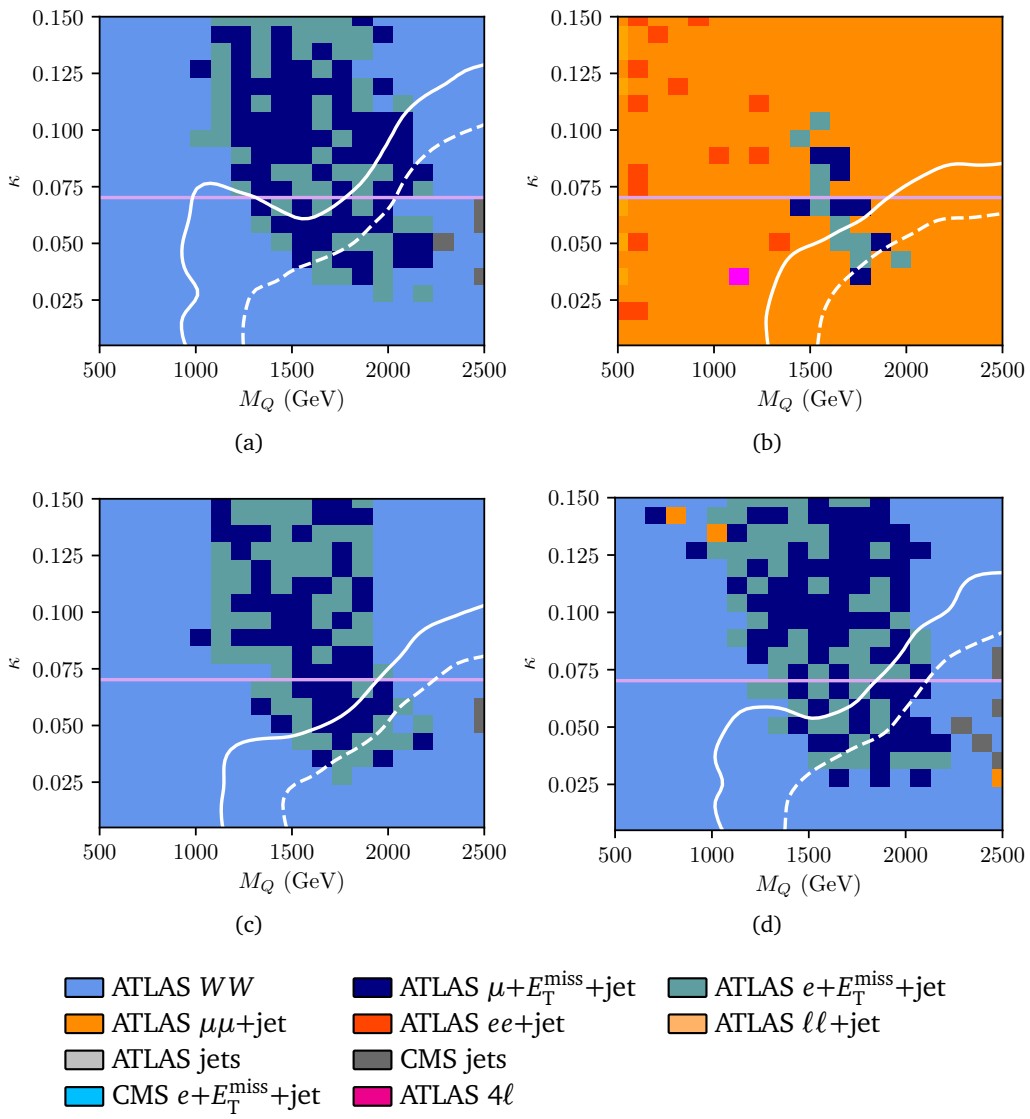

Figure 7: Dominant LHC analysis pools contributing to VLQ limit-setting in the $\kappa$ vs VLQ mass plane, where $\kappa$ is the coupling to first-generation SM quarks. All VLQ $(B,T,X,Y)$ masses are set to be degenerate. The disfavoured regions are located above and to the left of the dashed (68% CL) and solid (95% CL) white contours respectively. The lower bounds in $\kappa$ from non-LHC flavour physics are indicated with the pink horizontal contour. The VLQ branching fractions to $W$:$Z$:$H$ are (a) 0:0:1 (b) 0:1:0 (c) 1:0:0 and (d) $\frac{1}{2}$:$\frac{1}{4}$:$\frac{1}{4}$.

## 5.2 Second quark generation

In Figure 9, we perform the equivalent scan and analysis for weak VLQ couplings only to the second quark generation. Again, at low VLQ masses, pair production dominates and the exclusion is insensitive to $\kappa$. The same pattern of mass thresholds as a function of $\xi$ configuration is seen as for the first-generation scan. However, the $\kappa$-dependent exclusion which dominated the parameter space for the first generation is here absent, with only a hint of $\kappa$-dependence in the $Z$-only coupling configuration (Figure 9b). This is in keeping with the expectation summarised in Section 2, since the weak pair-production and single

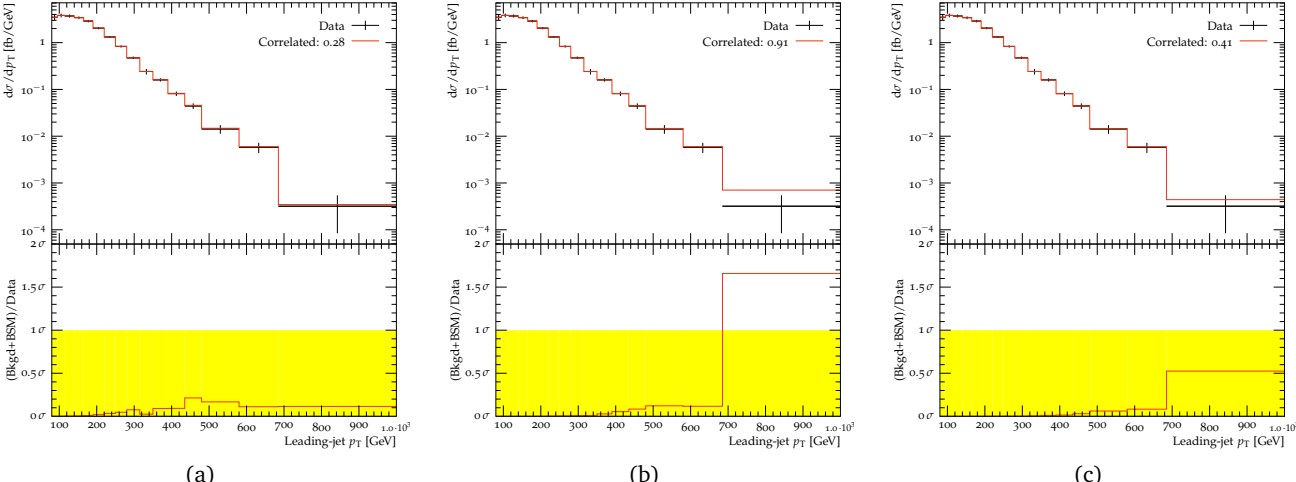

Figure 8: ATLAS 8 TeV $Wjj$ forward-lepton control region leading-jet $p_{\mathrm{T}}$ distributions at three points on the 95% exclusion contour for $W{:}Z{:}H = 1{:}0{:}0$, respectively at $M_Q$ values of (a) 1000 GeV, (b) 1750 GeV, and (c) 2250 GeV. The rise and subsidence of a 90% $\mathrm{CL_s}$ exclusion from a single $Wjj$ bin is seen as the contour passes from below 1 TeV to above 2 TeV. The black points are data, the red histogram is the VLQ contribution stacked on top of the data. In the lower insets, the ratio is shown and the yellow band indicates the significance, taking into account the statistical and systematic uncertainties on the data. The legend gives the exclusion (i.e. one minus the $p$-value) for that histogram after fitting nuisance parameters for the correlated systematic uncertainties.

production processes are highly suppressed if the VLQs cannot couple to the proton's valence quarks (nor the first-generation sea). In this case, the estimated limit ($\kappa < 0.2$) from non-LHC measurements comes from measurements of the $Z \to q\bar{q}$ couplings at LEP [4, 41]

The $WW$ analysis group dominates for the second generation $H$ and $W\xi$ corners (Figures 9a and 9c) and for most of the mixed-boson configuration, at least for all $\kappa$ values allowed by the flavour constraints. At higher $\kappa$ values, weak VLQ–boson production mechanisms again contribute, giving some weak dependence on the coupling. The $Z$-compatible dilepton+jets analyses dominate all of the $Z$-only configuration in Figure 9b. Thus for the most interesting $\kappa < 0.2$ region, second-generation limits are driven by VLQ pair production, with decays into the permitted bosons and SM quarks. This is consistent with the discussion in Section 2: production of VLQs in association to quarks is suppressed by an order of magnitude compared to the case where VLQs can couple to first-generation quarks, while pair-production via EM/QCD processes occurs at roughly the same rate. This explains the weaker dependence on $\kappa$ of the exclusion reported above.

We again see the intrusion of single-VLQ exclusion by the ATLAS VBF $Wjj$ analysis for second-generation VLQs between 1 TeV to 2 TeV, explained by the same mechanism. Compared to the first generation, however, the impact of QCD jet analyses is also seen at the highest masses, for all but the pure-$Z$ configuration of Figure 9b. This is primarily driven by the CMS 13 TeV jet mass analysis [42], and to a lesser extent the ATLAS 13 TeV dijet & inclusive jet analysis [43].

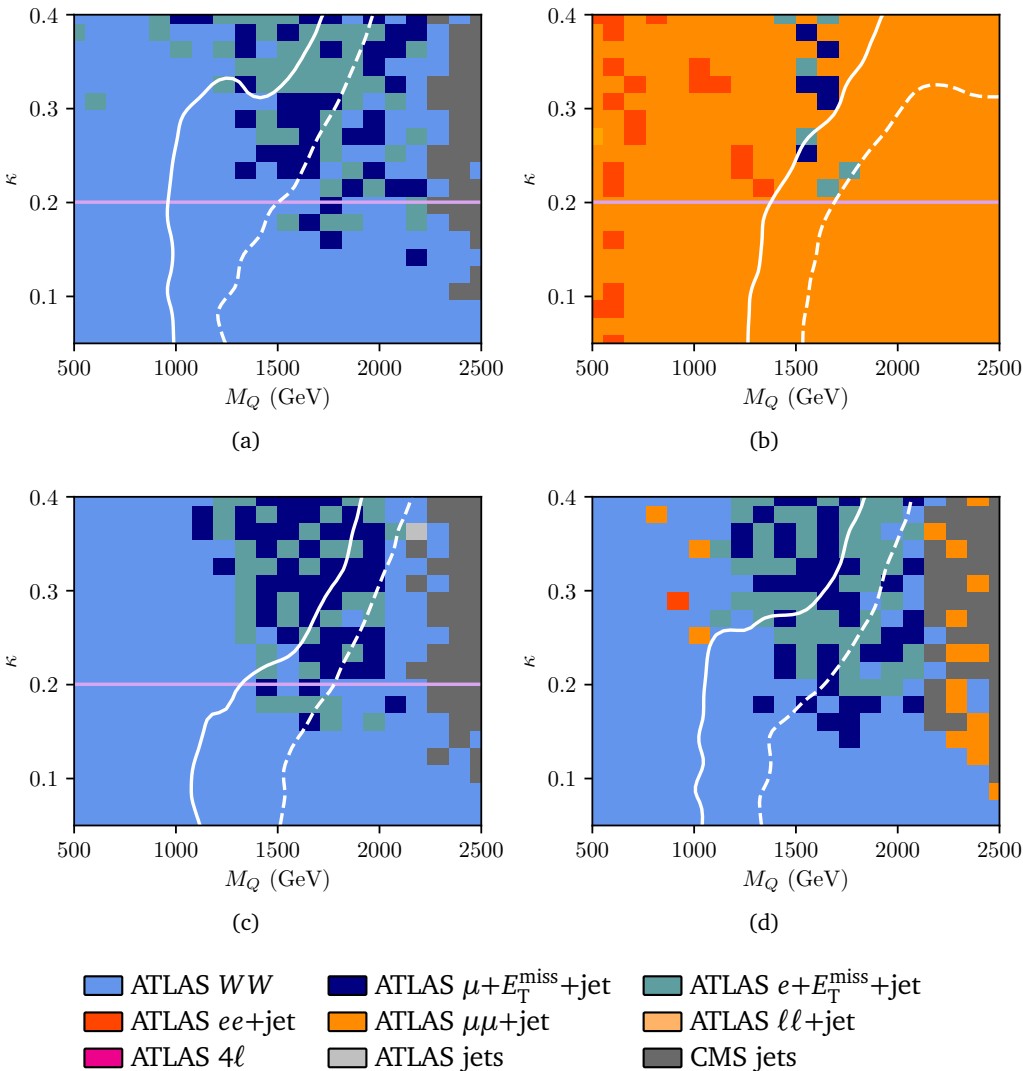

Figure 9: Dominant LHC analysis pools contributing to VLQ limit-setting in the $\kappa$ vs VLQ mass plane, where $\kappa$ is the coupling to second-generation SM quarks. All VLQ $(B, T, X, Y)$ masses are set to be degenerate. The disfavoured regions are located above and to the left of the dashed (68% CL) and solid (95% CL) white contours respectively. The lower bounds in $\kappa$ from non-LHC flavour physics are indicated with the pink horizontal contour. The VLQ branching fractions to $W$:$Z$:$H$ are (a) 0:0:1 (b) 0:1:0 (c) 1:0:0 and (d) $\frac{1}{2}$:$\frac{1}{4}$:$\frac{1}{4}$.

## 5.3 Third quark generation

In Figure 10, we show the sensitivity in the plane of the coupling $\kappa$ to the third-generation SM quarks and VLQ mass. Again, as discussed in Section 2, at low VLQ masses, pair production dominates and the exclusion is insensitive to $\kappa$. Since there is no top parton density, and the $b$ content of the proton is suppressed relative to light quarks, single production cross-sections are lower than for the lighter generations. Single production does still bring some additional sensitivity to higher masses (around 2 TeV) when $W + q$ decays dominate, which is consistent with the fact that in this region CONTUR does better than the searches (which focus on pair production) in Figure 5a (for $B$ VLQs).

Comparing Figure 10b with the equivalent figures for the first- and second-generation interactions (Figures 7b and 9b respectively), it is interesting to note that the type of measurement which provides the best sensitivity changes for the $Z$-corner in third-generation couplings. In the first- and second-generation cases, the sensitivity is dominated by dilepton-plus-jet measurements, while the third-generation case becomes dominated by measurements involving leptons and missing energy with jets, or $WW$-like measurements — only a few points in the scan are dominated by the ATLAS 13 TeV dilepton [29, 30] or four-lepton [44, 45] measurements.

This change is due to the fact that when third-generation couplings are the only ones allowed, $X + q$ and $T + q$ processes are suppressed due to the lack of top quarks in the proton sea. $Y + q$ dominates the high-mass region, with pair-production dominant for lower VLQ masses. The $T$ and $Y$ VLQs, produced singly or in pairs, will decay to top quarks, resulting in the production of at least one $W$-boson. This in turn leads to the missing energy signatures which were not present for first- or second-generation couplings.

The $WW$ dominance across all $\xi$ configurations also reflects the impact of published bin-correlation data: without this information, permitting simultaneous use of multiple bins in the $WW$ analysis pool, the 13 TeV CMS $\ell + E_T^{\text{miss}}$+jet analyses [33–35, 46–48] would dominate at low mass in the non-$Z$ configurations, and the statistically limited ATLAS four-lepton analyses would dominate a larger region of the $\xi^Z = 1$ $\kappa$–$M_Q$ plane.

## 5.4 Singlets, doublets and triplets

The results in this section assumed four VLQs, $B, T, X, Y$. This was a good didactic scenario, as it contains the most rich phenomenology, but is not the most favoured scenario. In Appendix A we present equivalent results for singlets $(B)$, $(T)$, doublets $(B, T)$, $(B, Y)$, $(X, T)$, and triplets $(B, T, X)$, $(B, T, Y)$, which have stronger theoretical motivation. The $W{:}Z{:}H = 0{:}\frac{1}{2}{:}\frac{1}{2}$ case is considered for doublets instead of $W{:}Z{:}H = \frac{1}{2}{:}\frac{1}{4}{:}\frac{1}{4}$, as motivated by Ref. [4]. The results are broadly similar to those shown in the previous sections, with a few important differences which we summarise here:

- The constraints are typically slightly weaker the fewer particles there are in the multiplet: more new particles means more opportunities for divergences from the SM.

- Scenarios without $X$ or $Y$ particles and where $W{:}Z{:}H = 0{:}0{:}1$ have substantially weaker constraints than other scenarios, since the lack of $W$ decays means that only Higgs measurements would be sensitive (manifested as $\gamma + X$ analyses dominating low-mass sensitivity via $H \to \gamma\gamma$), and low single-production cross-sections through the Higgs mean that only comparatively high values of $\kappa$ can be excluded beyond the pair-production region.

- Continuing this focus on the $W{:}Z{:}H = 0{:}0{:}1$ configuration, if there are only first-generation couplings, the triplets and doublets containing only one of $X$ or $Y$ have differing degrees of $\kappa$-sensitive exclusion power, with stronger constraints if $X$ is present. This is due to the larger valence up- versus down-quark PDFs, for $u \to W^- X$ and $d \to W^+ Y$ respectively. This effect disappears (along with the valence PDFs) for second-generation couplings.

- Assuming only third-generation couplings, the $W{:}Z{:}H = 0{:}1{:}0$ and $W{:}Z{:}H = 0{:}0{:}1$ corners for multiplets containing only $T$ and $Y$ VLQs have no $\kappa$-dependence in their constraints, since single-VLQ production is not allowed: the top proton density is vanishingly small for $t \to T\{Z, H\}$ and $t \to XW^-$ production. Only pair-production is a viable mechanism.

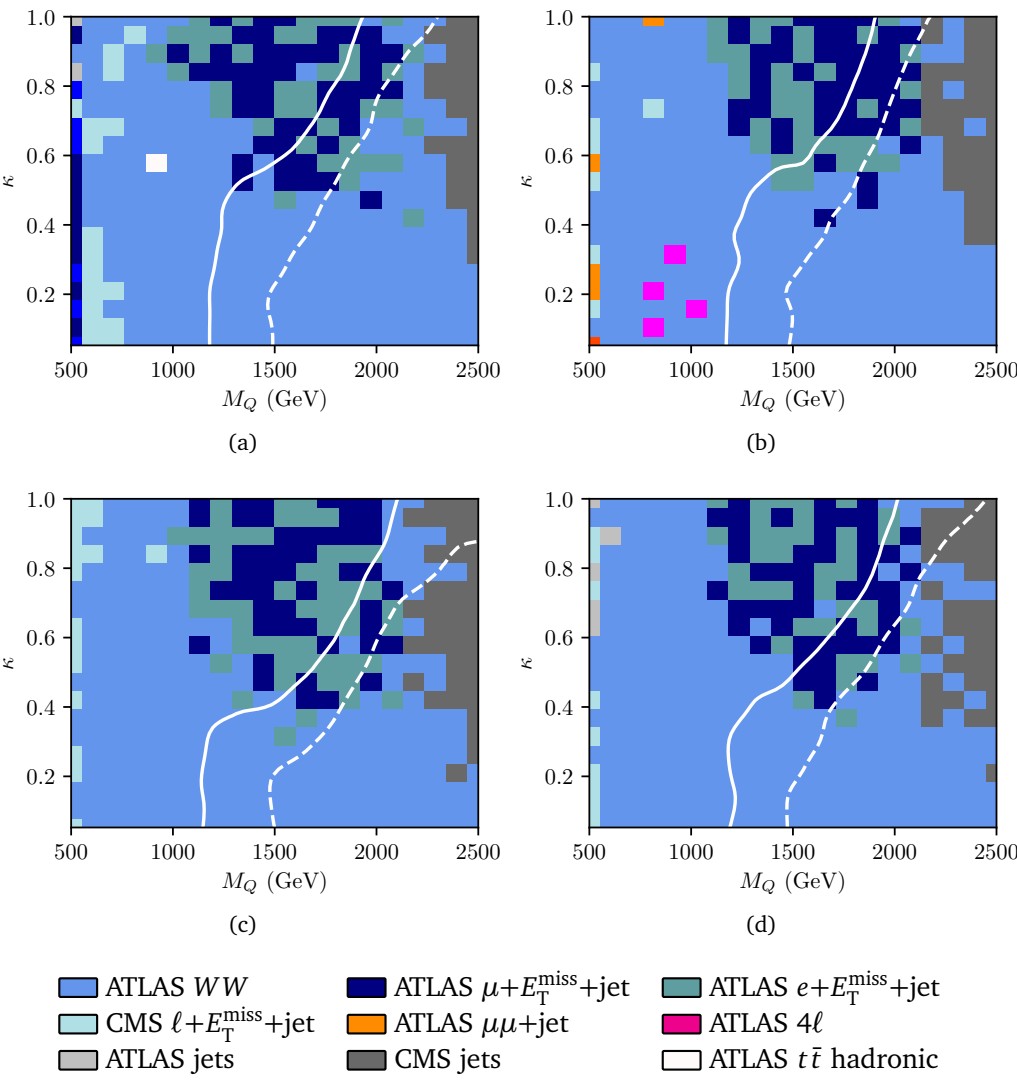

Figure 10: Dominant LHC analysis pools contributing to VLQ limit-setting in the $\kappa$ vs VLQ mass plane, where $\kappa$ is the coupling to third-generation SM quarks. All VLQ $(B, T, X, Y)$ masses are set to be degenerate. The disfavoured regions are located above and to the left of the dashed (68% CL) and solid (95% CL) white contours respectively. The VLQ branching fractions to $W$:$Z$:$H$ are (a) 0:0:1 (b) 0:1:0 (c) 1:0:0 and (d) $\frac{1}{2}$:$\frac{1}{4}$:$\frac{1}{4}$.

- The converse would apply for only $B$ and $X$ VLQs in the $W$:$Z$:$H$ = 1:0:0 case, but the only such natural multiplet is the $B$ singlet. In this we do indeed see $\kappa$ independence again.

# 6 Discussion and conclusions

We have presented studies of a generic class of Vector-Like Quark models, comparing the predictions from all $2 \rightarrow 2$ production diagrams for VLQs to a large bank of LHC differential cross-section measurements. Despite the measurements not being explicitly designed for this purpose, we find that they can exclude significant regions of VLQ parameter space, in a wider

range of model parameters than those typically considered in dedicated searches.

This approach, using inclusive event generation and model-independent measurements to study BSM signals, is hence not only competitive with dedicated searches, but indeed can outperform them in generic model spaces for which specific searches have not been optimised. This is due to the fact that searches often make simplifying assumptions, or focus their efforts on detecting the most spectacular signature from a model. But studying all the implications of a given model simultaneously may instead reveal moderate changes to many SM distributions, an effect most evident through combination of multiple precision measurements. This is very much the case in VLQ models. Firstly, LHC searches for such particles have often focused on pair-produced VLQs as a $\kappa$-dependence is technically challenging for interpretations made using detector-level physics objects. Secondly, it is typical at the LHC for only VLQs coupling to third-generation quarks to be considered, as they give the most striking experimental signatures and are most weakly constrained by previous measurements. Thirdly we note efforts to reinterpret searches in terms of models that include additional exotic VLQ production mechanisms [49]; our approach could be easily extended to cover such scenarios.

Reviewing the phenomenology of VLQ models at the LHC, we see that the way in which VLQs are produced in proton–proton collisions depends intimately on interplays between the composition of the proton, and the strengths of the coupling of VLQs to different SM bosons and the different generations of SM quarks. For example, VLQs coupled to first-generation quarks are more likely to be singly-produced than if coupled to second- or third-generation quarks, where pair production is typically more likely. Furthermore, production mechanisms such as $X + W$ become almost forbidden if only third-generation couplings are allowed, leading to an asymmetry between $X$ and $Y$ production which is not present for second-generation couplings. Finally, the phenomenology of VLQ decays when $B, T$ only couple to $Z$ bosons changes drastically if VLQs are allowed to couple to third-generation quarks: lepton-plus-missing-energy signatures then dominate over the expected dilepton-plus-jet signatures, due to the production of top quarks and subsequent decays involving missing energy.

These effects are neither small nor trivial, and suggest that the richly intertwined phenomenology of VLQ production and decay at the LHC could lead to novel analysis strategies and potentially new routes to discovery at hadron colliders. Indeed, if a discovery were made, one could use these considerations, along with the latest understanding of proton PDFs, to constrain both the overall scale of weak VLQ interactions and their relative couplings to different SM quark generations. These insights from our inclusive approach herald a very interesting era for VLQ searches at the LHC.

Given that a dedicated search can take a large team several years to prepare and publish, while running a CONTUR scan takes less than a day, it is arguable that checking compatibility of the search targets with the current canon of model-independent measurements constitutes an important "due diligence" step in analysis design. The increased volume of data, and pressures on computing resources in future LHC runs, make this argument ever more compelling in our view. Conversely, actively considering the contributions of inclusive-production studies when designing future analyses enables analysis teams to focus on those model regions which are not already covered.

This would naturally focus search attention on more exotic signatures, for example of long-lived particles or other anomalous, intrinsically non-SM-like features. In the precision era of the LHC there is an increasing motivation to make detailed measurements, rather than searches, in regions of phase-space with a significant SM cross-section. Search analyses can also contribute to the resource of model-independent results by making unfolded measurements in their control regions[3]: as discussed in this paper, a prototype of control-region measurements in a leptoquark

---

[3]And in signal regions if no excess is observed, both adding re-interpretation value and enhancing the impact of an otherwise null-result paper.

search has proven to have significant exclusion power in regions of VLQ parameter space. One search's control region can be anothers signal region: model-independent measurements may help avoid the danger of accidentally fitting away a signal.

Finally, this approach relies on careful preservation of measurements in HEPData and RIVET. Although RIVET was designed as a way to compare MC generators, it turns out to be perfectly suited to re-interpretation, especially if the reference data are published with a full breakdown of uncertainties and state-of-the-art background predictions to aid rigorous statistical interpretation. Most LHC measurements do provide RIVET routines, but unfortunately some very powerful measurements are still missing from the RIVET database, or only come out many years after the associated publication. Furthermore, CMS has historically published fewer routines than ATLAS, a fact manifest in the dominance by ATLAS measurements of the $CL_s$ limits shown in this paper. We hope that our results add further encouragement for all collider physics experiments to make public re-interpretation routines and bin-correlation data a core feature of their publication and data-preservation processes [50].

# Acknowledgements

We thank Jack Burton, Khadeeja Bepari, Ben Waugh and David Yallup for helpful discussions, Martin Habedank for implementing the presentation of dominant $CL_s$ pools in CONTUR, and Peter Richardson for improvements and enhancements of the Herwig UFO interface. Our thanks to the many contributors to the RIVET analysis collection.

**Funding information** AB, JMB, LC and DH have received funding from the European Union's Horizon 2020 research and innovation programme as part of the Marie Skłodowska-Curie Innovative Training Network MCnetITN3 (grant agreement no. 722104). AB and PS acknowledge Royal Society funding under grants UF160548 and RGF\EA\180252. AB, JMB and LC have received funding from the UKRI Science and Technology Facilities Council (STFC) consolidated grants for experimental particle physics.

# A Additional VLQ multiplets

## A.1 First generation

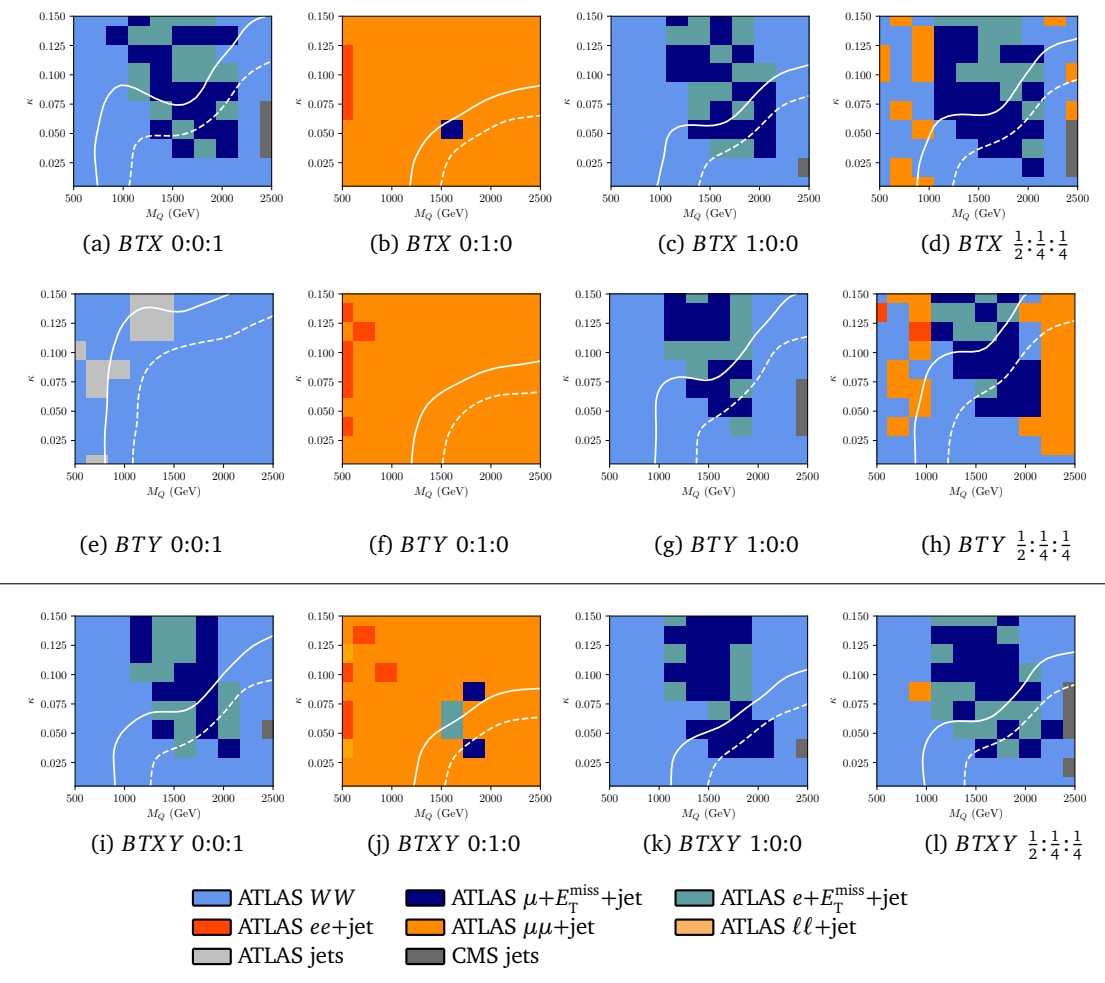

Figure 11: Sensitivity of LHC measurements to VLQ triplet production in the $\kappa$ vs VLQ mass plane, where $\kappa$ is the coupling to first-generation SM quarks. All VLQ masses are set to be degenerate. The multiplets are given as rows: (a)–(d) $(B, T, X)$ triplet, (e)–(h) $(B, T, Y)$ triplet, and for comparison to the main text, the (i)–(l) $(B, T, X, Y)$ quadruplet. The VLQ branching fractions to $W{:}Z{:}H$ are arranged in columns of 0:0:1, 0:1:0, 1:0:0, and $\frac{1}{2}{:}\frac{1}{4}{:}\frac{1}{4}$ from left to right.

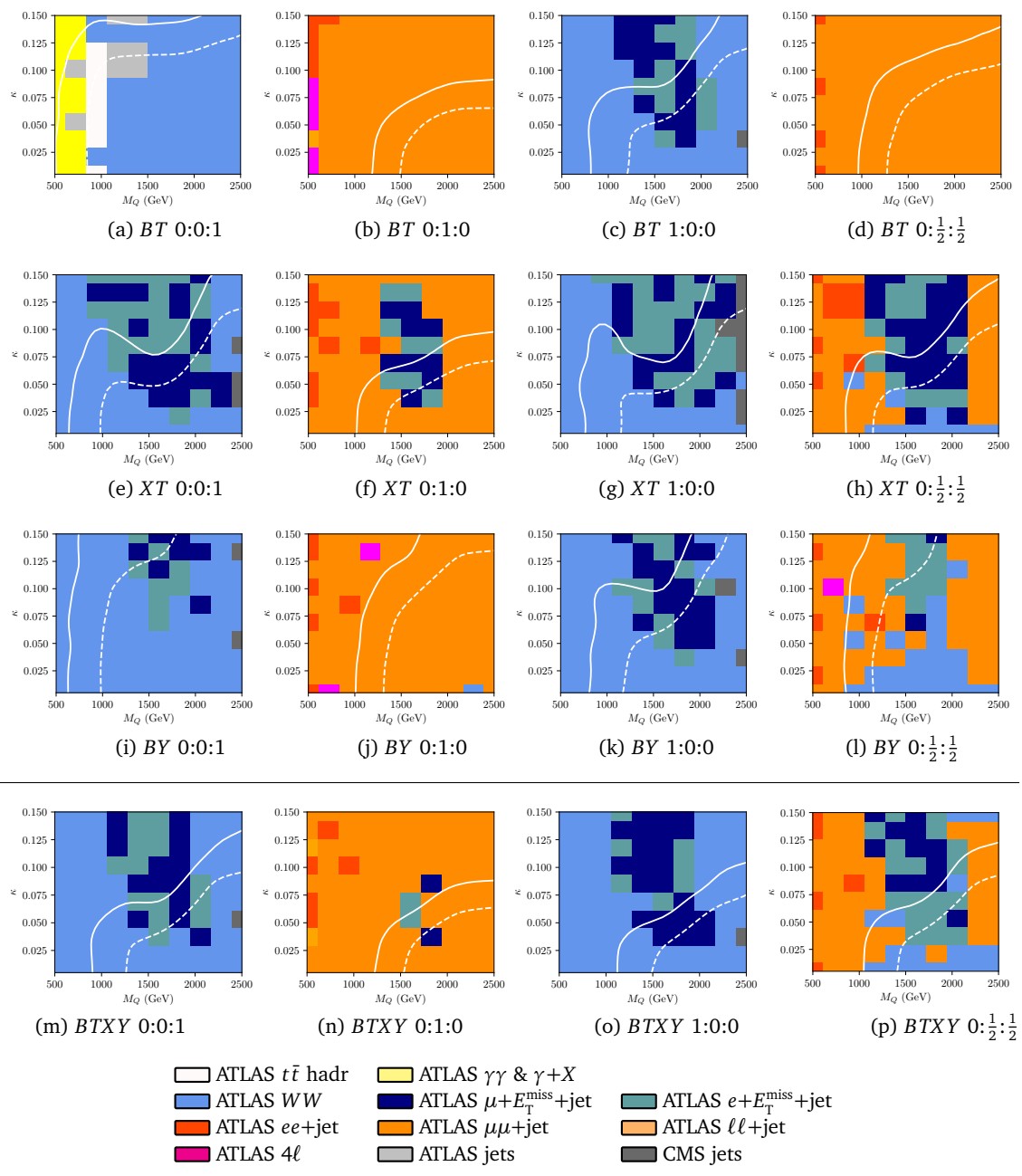

Figure 12: Sensitivity of LHC measurements to VLQ doublet production in the $\kappa$ vs VLQ mass plane, where $\kappa$ is the coupling to first-generation SM quarks. All VLQ masses are set to be degenerate. The multiplets are given as rows: (a)–(d) $(B, T)$ doublet, (e)–(h) $(X, T)$ doublet, (i)–(l) $(B, Y)$ doublet, and for comparison to the main text, the (m)–(o) $(B, T, X, Y)$ quadruplet. The VLQ branching fractions to $W$:$Z$:$H$ are arranged in columns of 0:0:1, 0:1:0, 1:0:0, and 0:$\frac{1}{2}$:$\frac{1}{2}$ from left to right. The 0:$\frac{1}{2}$:$\frac{1}{2}$ case is considered for doublets instead of $\frac{1}{2}$:$\frac{1}{4}$:$\frac{1}{4}$, as motivated by Ref. [4].

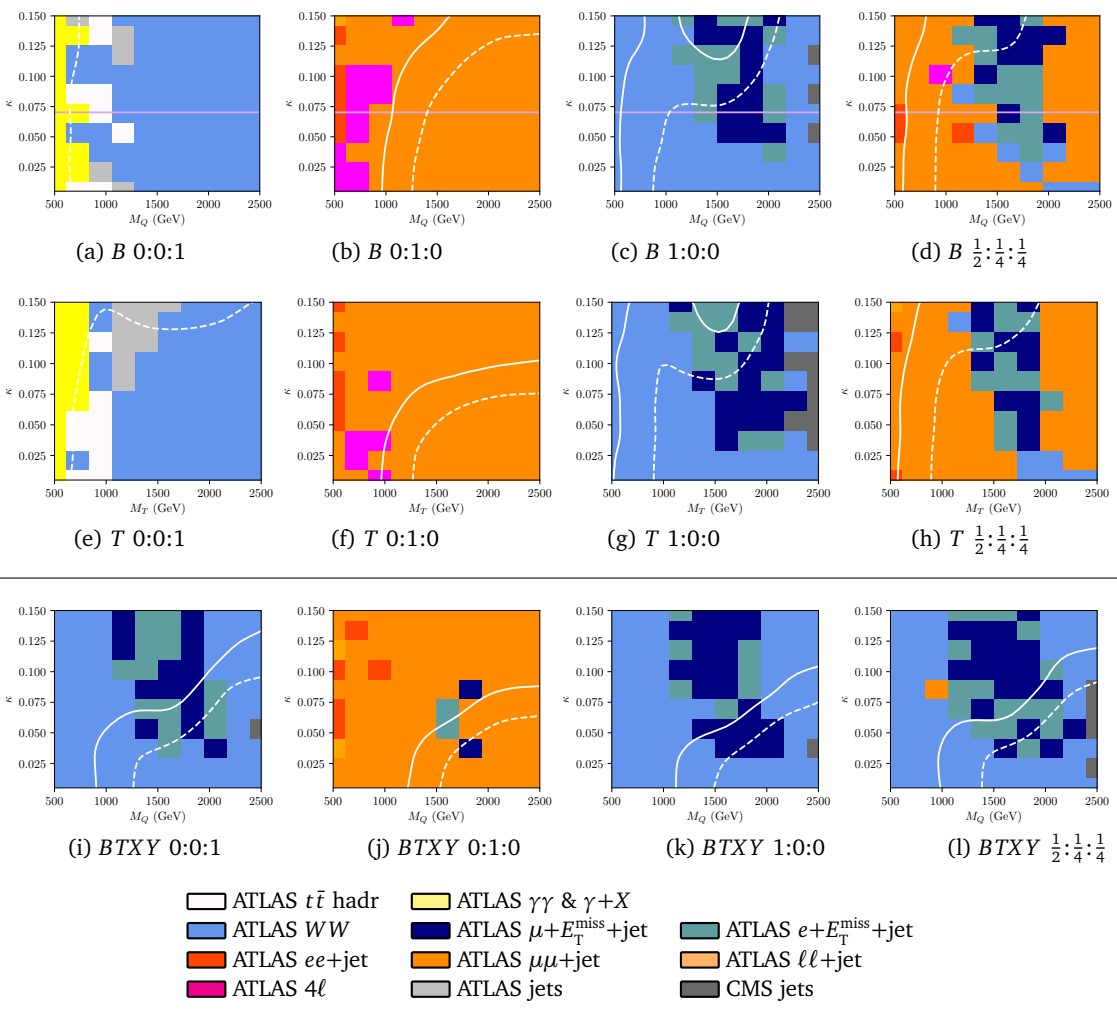

Figure 13: Sensitivity of LHC measurements to VLQ singlet production in the $\kappa$ vs VLQ mass plane, where $\kappa$ is the coupling to first-generation SM quarks. All VLQ masses are set to be degenerate. The multiplets are given as rows: (a)–(d) $B$ singlet, (e)–(h) $T$ singlet, and for comparison to the main text, the (i)–(l) $(B, T, X, Y)$ quadruplet. The VLQ branching fractions to $W$:$Z$:$H$ are arranged in columns of 0:0:1, 0:1:0, 1:0:0, and $\frac{1}{2}$:$\frac{1}{4}$:$\frac{1}{4}$ from left to right.

## A.2 Second generation

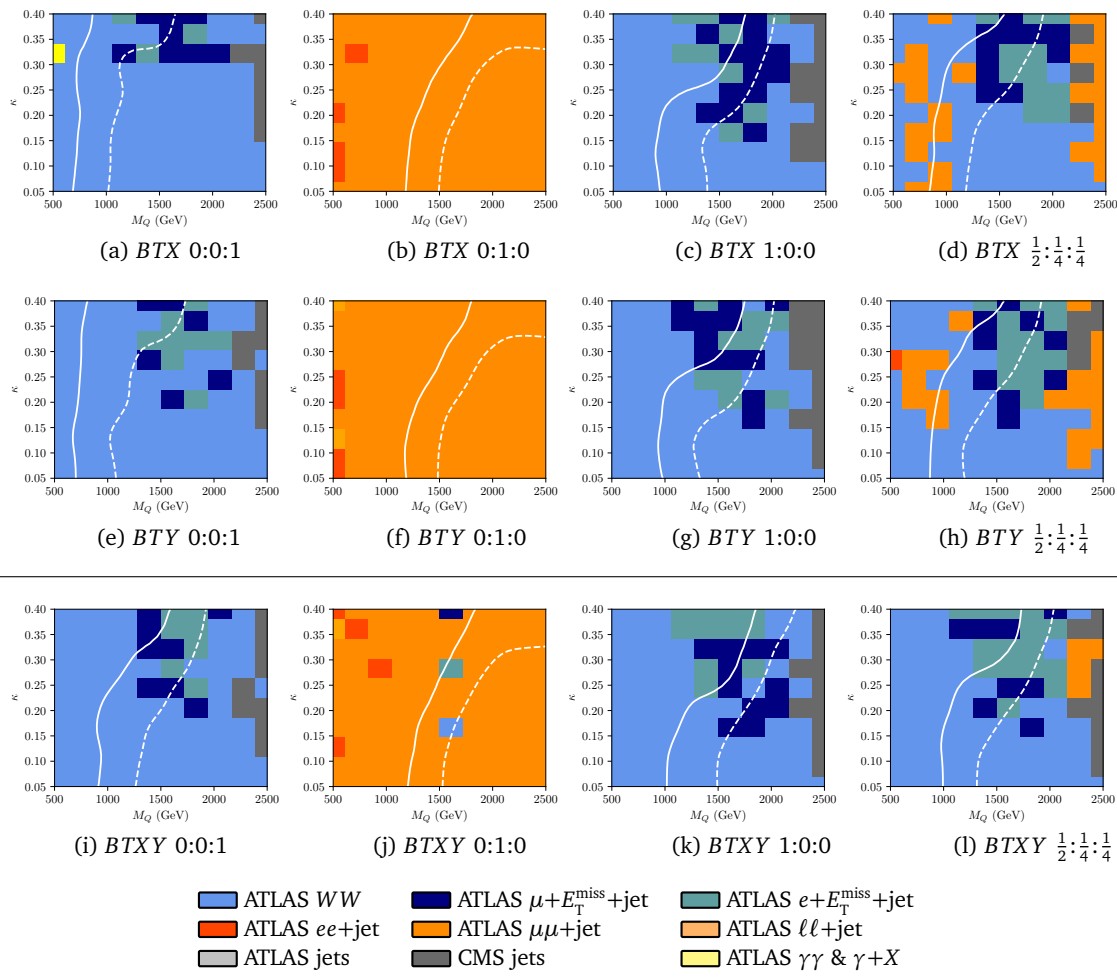

Figure 14: Sensitivity of LHC measurements to VLQ triplet production in the $\kappa$ vs VLQ mass plane, where $\kappa$ is the coupling to second-generation SM quarks. All VLQ masses are set to be degenerate. The multiplets are given as rows: (a)–(d) $(B, T, X)$ triplet, (e)–(h) $(B, T, Y)$ triplet, and for comparison to the main text, the (i)–(l) $(B, T, X, Y)$ quadruplet. The VLQ branching fractions to $W$:$Z$:$H$ are arranged in columns of 0:0:1, 0:1:0, 1:0:0, and $\frac{1}{2}$:$\frac{1}{4}$:$\frac{1}{4}$ from left to right.

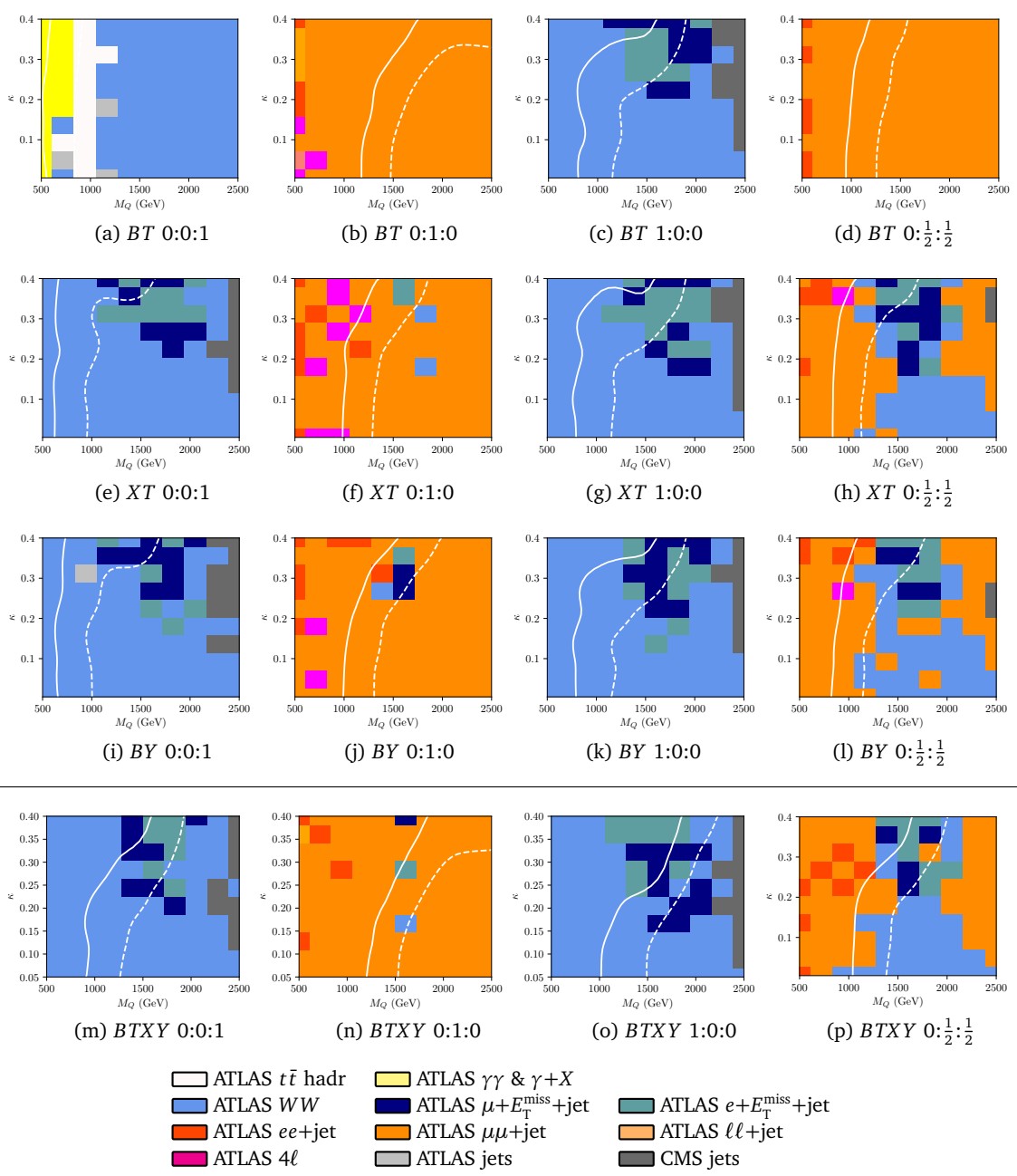

Figure 15: Sensitivity of LHC measurements to VLQ doublet production in the $\kappa$ vs VLQ mass plane, where $\kappa$ is the coupling to second-generation SM quarks. All VLQ masses are set to be degenerate. The multiplets are given as rows: (a)–(d) $(B, T)$ doublet, (e)–(h) $(X, T)$ doublet, (i)–(l) $(B, Y)$ doublet, and for comparison to the main text, the (m)–(o) $(B, T, X, Y)$ quadruplet. The VLQ branching fractions to $W$:$Z$:$H$ are arranged in columns of 0:0:1, 0:1:0, 1:0:0, and 0:$\frac{1}{2}$:$\frac{1}{2}$ from left to right. The 0:$\frac{1}{2}$:$\frac{1}{2}$ case is considered for doublets instead of $\frac{1}{2}$:$\frac{1}{4}$:$\frac{1}{4}$, as motivated by Ref. [4].

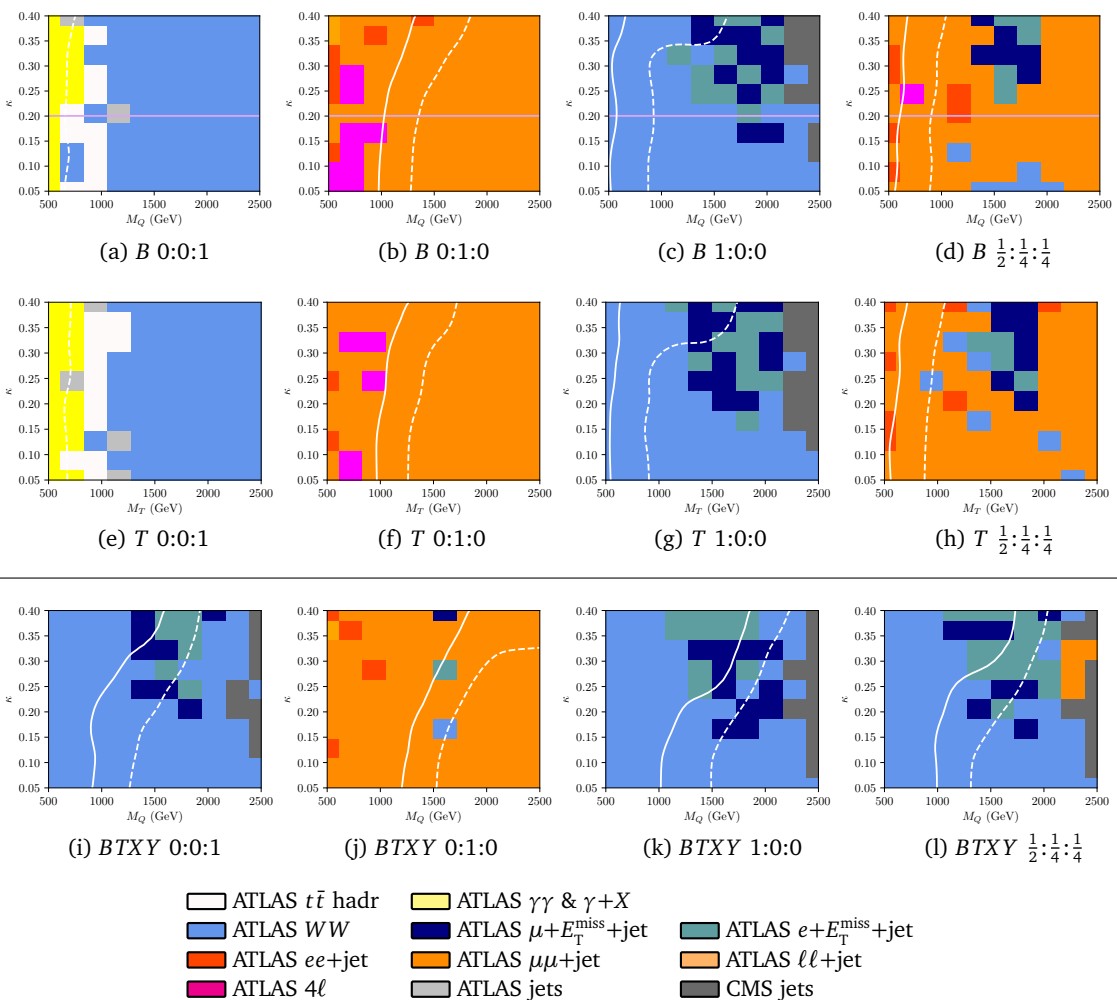

Figure 16: Sensitivity of LHC measurements to VLQ singlet production in the $\kappa$ vs VLQ mass plane, where $\kappa$ is the coupling to second-generation SM quarks. All VLQ masses are set to be degenerate. The multiplets are given as rows: (a)–(d) $B$ singlet, (e)–(h) $T$ singlet, and for comparison to the main text, the (i)–(l) $(B, T, X, Y)$ quadruplet. The VLQ branching fractions to $W$:$Z$:$H$ are arranged in columns of 0:0:1, 0:1:0, 1:0:0, and $\frac{1}{2}$:$\frac{1}{4}$:$\frac{1}{4}$ from left to right.

## A.3 Third generation

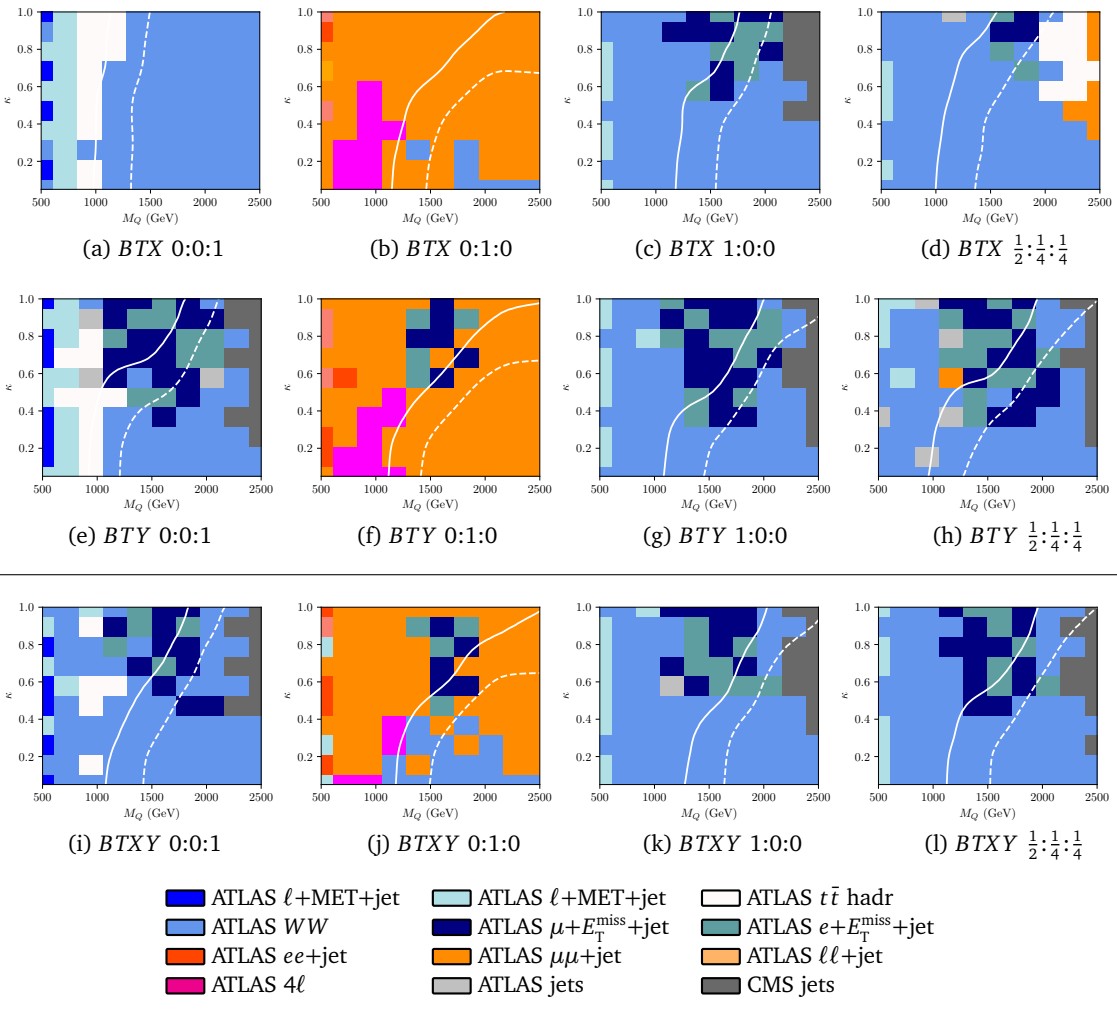

Figure 17: Sensitivity of LHC measurements to VLQ triplet production in the $\kappa$ vs VLQ mass plane, where $\kappa$ is the coupling to third-generation SM quarks. All VLQ masses are set to be degenerate. The multiplets are given as rows: (a)–(d) $(B, T, X)$ triplet, (e)–(h) $(B, T, Y)$ triplet, and for comparison to the main text, the (i)–(l) $(B, T, X, Y)$ quadruplet. The VLQ branching fractions to $W$:$Z$:$H$ are arranged in columns of 0:0:1, 0:1:0, 1:0:0, and $\frac{1}{2}$:$\frac{1}{4}$:$\frac{1}{4}$ from left to right.

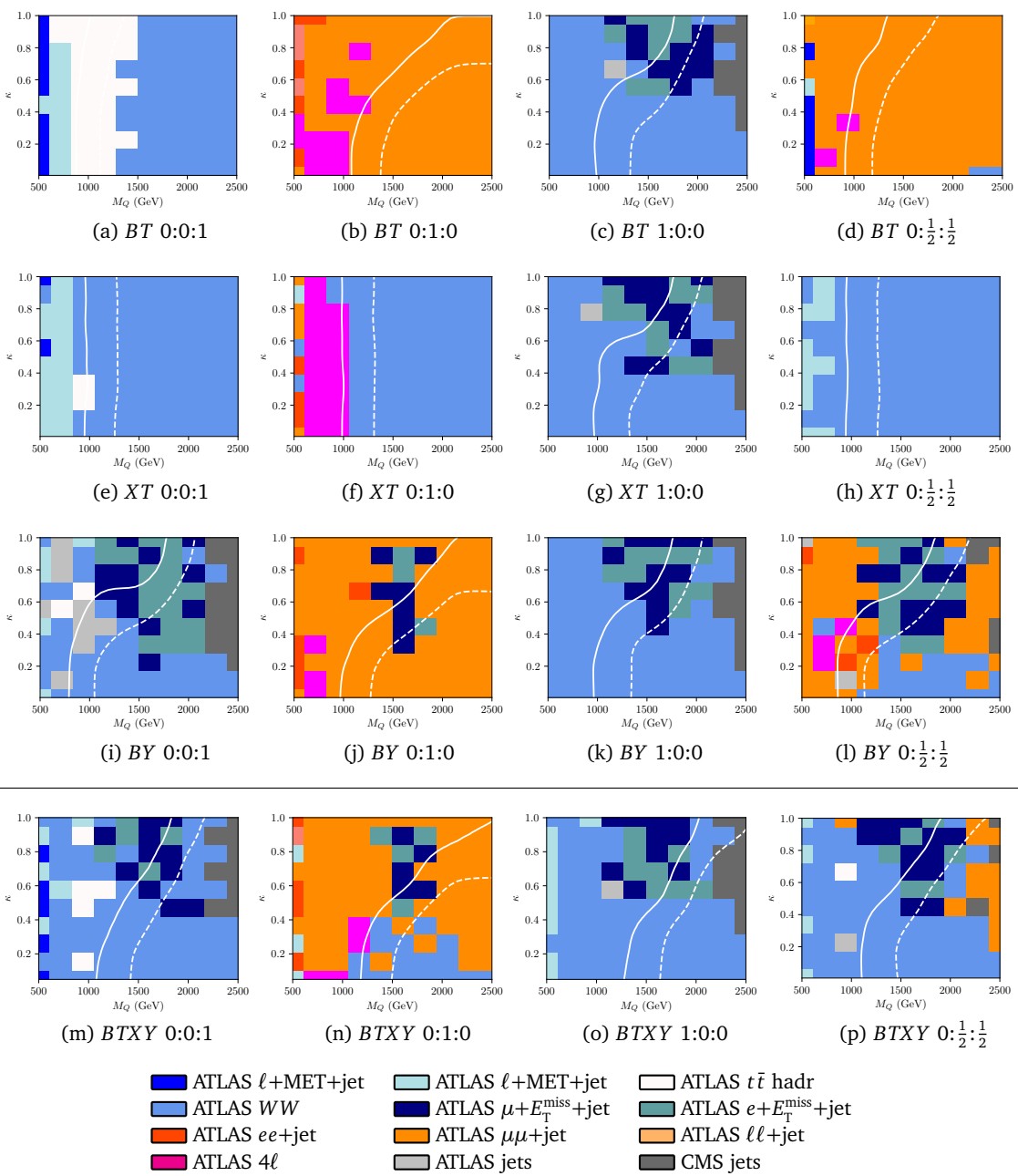

Figure 18: Sensitivity of LHC measurements to VLQ doublet production in the $\kappa$ vs VLQ mass plane, where $\kappa$ is the coupling to third-generation SM quarks. All VLQ masses are set to be degenerate. The multiplets are given as rows: (a)–(d)$(B,T)$ doublet, (e)–(h)$(X,T)$ doublet, (i)–(l)$(B,Y)$ doublet, and for comparison to the main text, the (m)–(o)$(B,T,X,Y)$ quadruplet. The VLQ branching fractions to $W$:$Z$:$H$ are arranged in columns of 0:0:1, 0:1:0, 1:0:0, and 0:$\frac{1}{2}$:$\frac{1}{2}$ from left to right. The 0:$\frac{1}{2}$:$\frac{1}{2}$ case is considered for doublets instead of $\frac{1}{2}$:$\frac{1}{4}$:$\frac{1}{4}$, as motivated by Ref. [4].

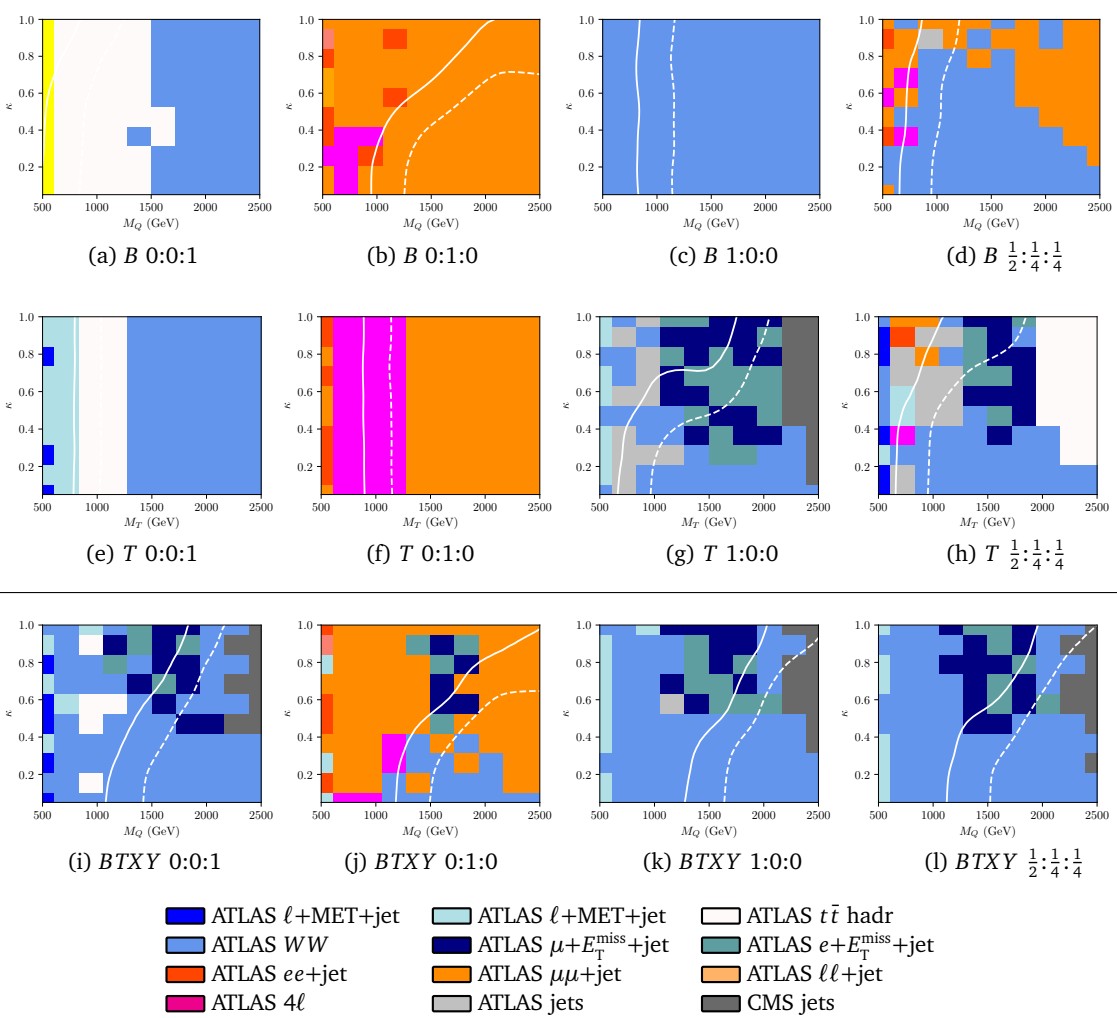

Figure 19: Sensitivity of LHC measurements to VLQ singlet production in the $\kappa$ vs VLQ mass plane, where $\kappa$ is the coupling to third-generation SM quarks. All VLQ masses are set to be degenerate. The multiplets are given as rows: (a)–(d) $B$ singlet, (e)–(h) $T$ singlet, and for comparison to the main text, the (i)–(l) $(B, T, X, Y)$ quadruplet. The VLQ branching fractions to $W$:$Z$:$H$ are arranged in columns of 0:0:1, 0:1:0, 1:0:0, and $\frac{1}{2}$:$\frac{1}{4}$:$\frac{1}{4}$ from left to right.

# B CL$_s$ maps in $\kappa$ vs $M_Q$

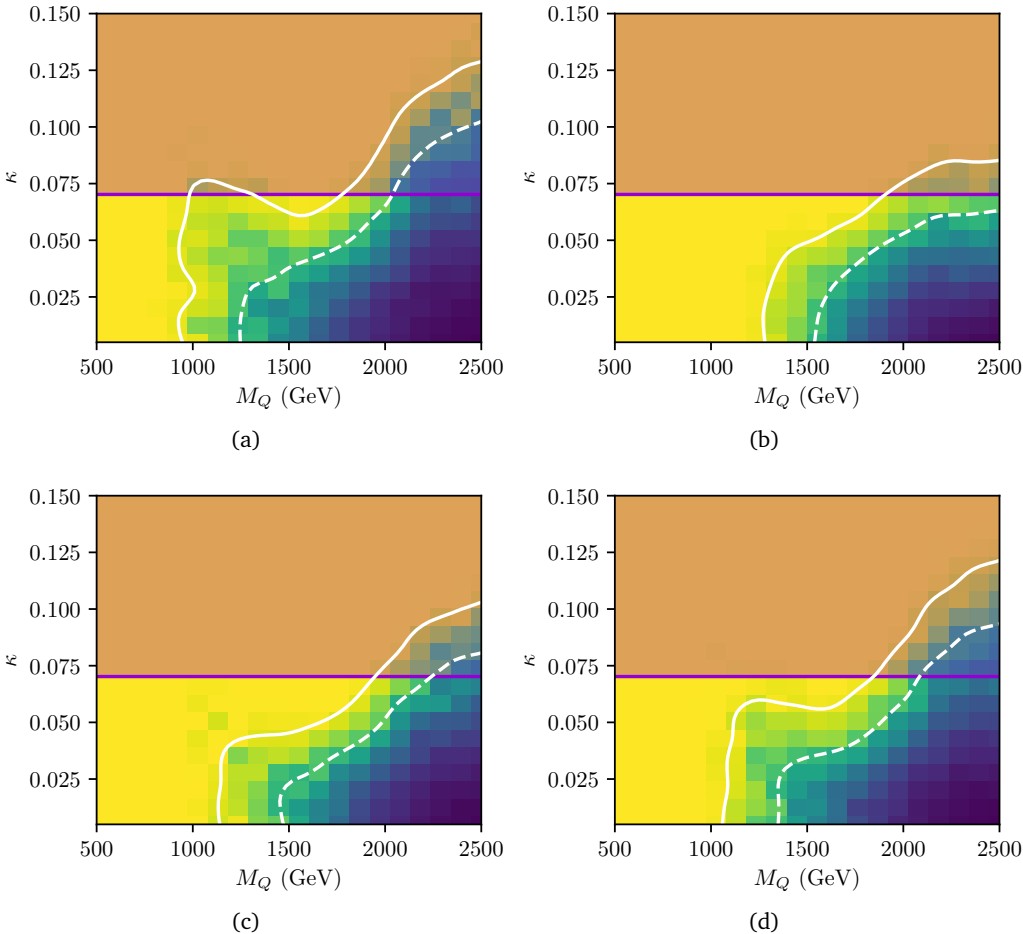

Figure 20: Sensitivity of LHC measurements to VLQ production in the $\kappa$ vs VLQ mass plane, where $\kappa$ is the coupling to first-generation SM quarks. All VLQ ($B, T, X, Y$) masses are set to be degenerate. The green and yellow regions are disfavoured at 68% CL and 95% CL respectively, with the dashed and solid white contours delineating the boundaries. The VLQ branching fractions to $W$:$Z$:$H$ are (a) 0:0:1 (b) 0:1:0 (c) 1:0:0 and (d) $\frac{1}{2}$:$\frac{1}{4}$:$\frac{1}{4}$.

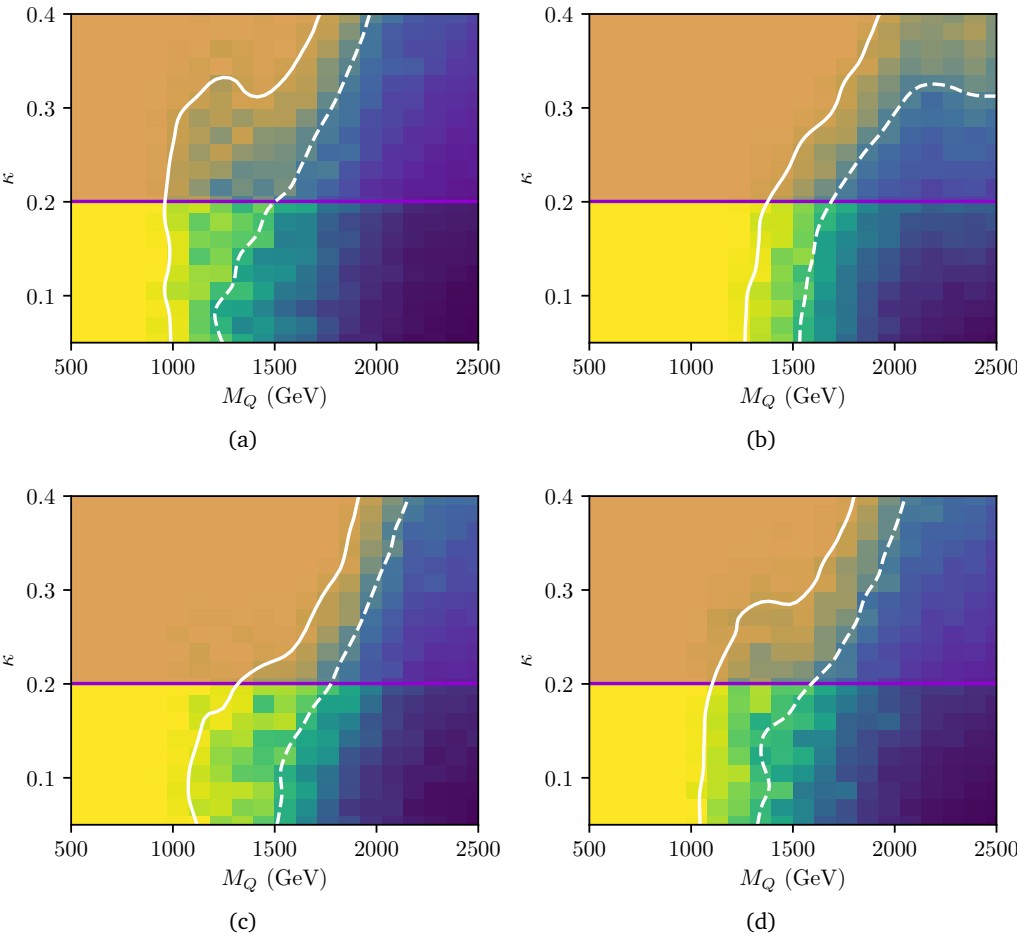

Figure 21: Sensitivity of LHC measurements to VLQ production in the $\kappa$ vs VLQ mass plane, where $\kappa$ is the coupling to second-generation SM quarks. All VLQ $(B, T, X, Y)$ masses are set to be degenerate. The green and yellow regions are disfavoured at 68% CL and 95% CL respectively, with the dashed and solid white contours delineating the boundaries. The VLQ branching fractions to $W{:}Z{:}H$ are (a) 0:0:1 (b) 0:1:0 (c) 1:0:0 and (d) $\frac{1}{2}{:}\frac{1}{4}{:}\frac{1}{4}$.

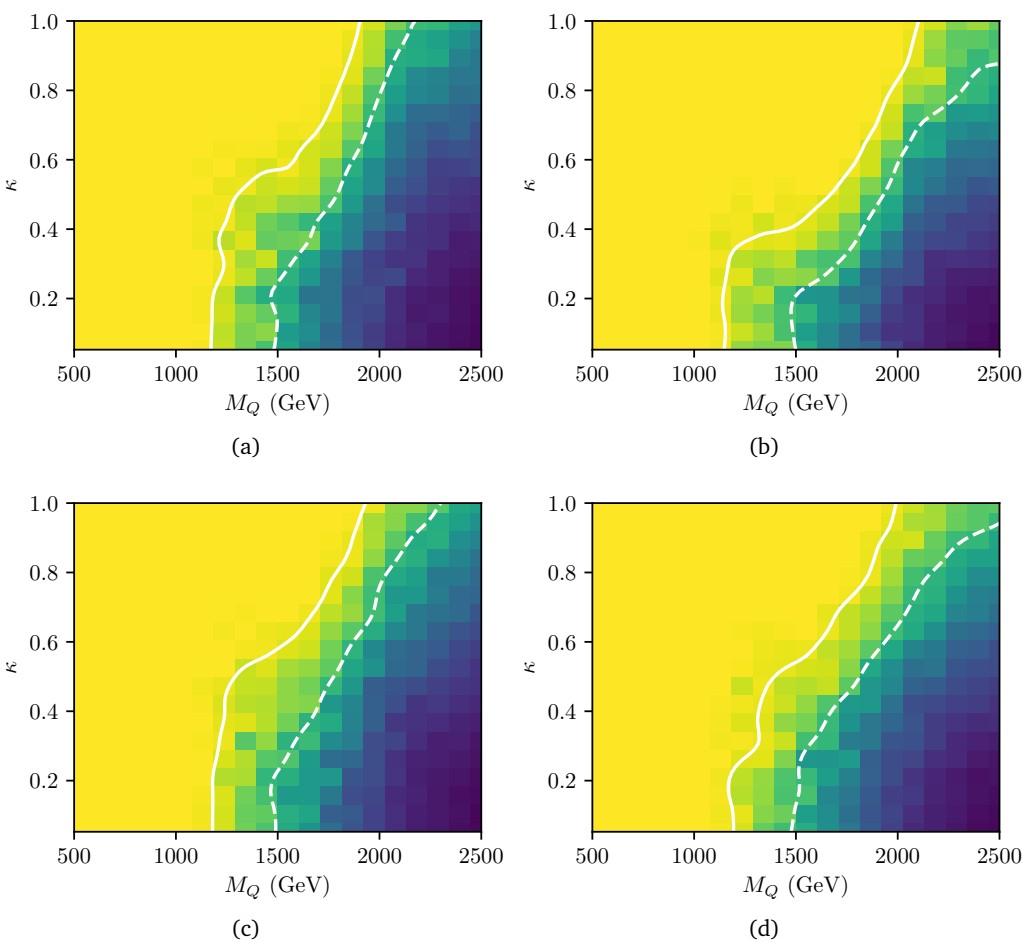

Figure 22: Sensitivity of LHC measurements to VLQ production in the $\kappa$ vs VLQ mass plane, where $\kappa$ is the coupling to third-generation SM quarks. All VLQ $(B, T, X, Y)$ masses are set to be degenerate. The green to yellow regions are disfavoured at 68% CL to 95% CL respectively, with the dashed and solid white contours delineating the boundaries. The VLQ branching fractions to $W{:}Z{:}H$ are (a) 0:0:1 (b) 0:1:0 (c) 1:0:0 and (d) $\frac{1}{2}{:}\frac{1}{4}{:}\frac{1}{4}$.

# C   Uncorrelated 3rd generation dominant-analyses maps

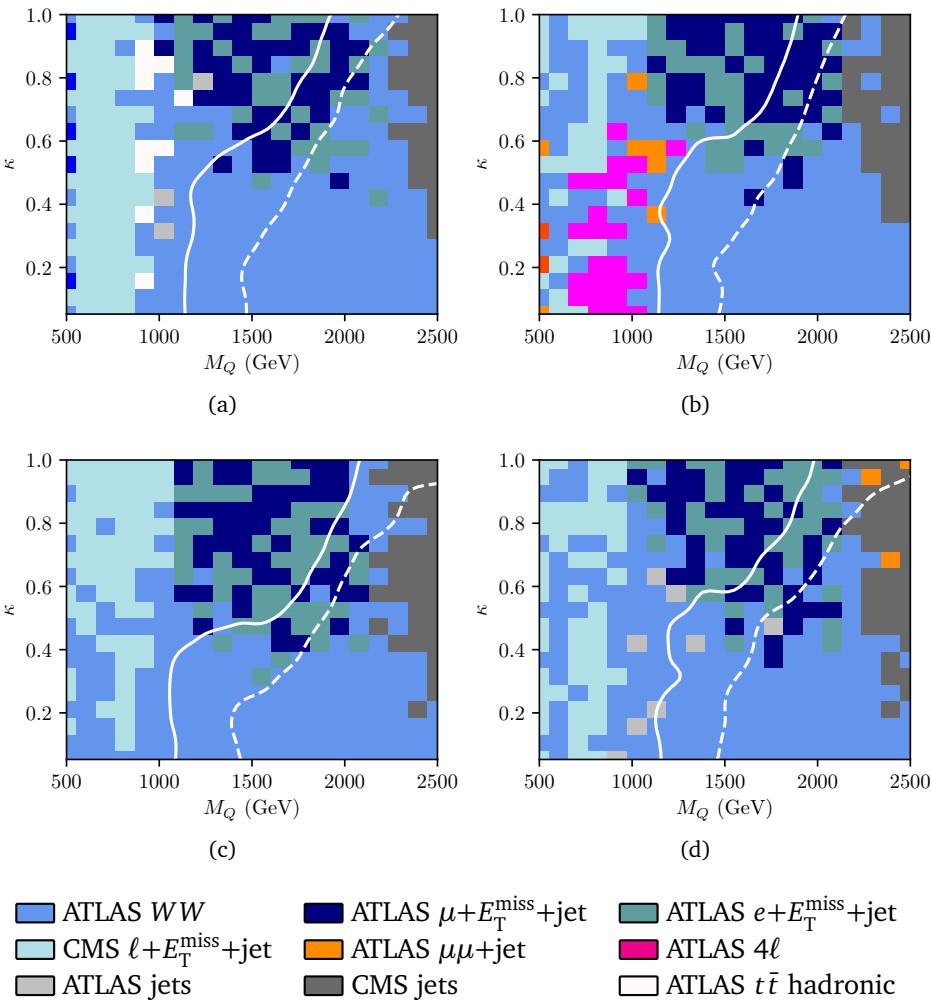

Figure 23: Dominant LHC analysis pools contributing to VLQ limit-setting in the $\kappa$ vs VLQ mass plane, where $\kappa$ is the coupling to third-generation SM quarks, *without* bin-to-bin correlations included in the $\mathrm{CL_s}$ calculations. All VLQ $(B, T, X, Y)$ masses are set to be degenerate. The disfavoured regions are located above and to the left of the dashed (68% CL) and solid (95% CL) white contours respectively. The VLQ branching fractions to *W*:*Z*:*H* are (a) 0:0:1 (b) 0:1:0 (c) 1:0:0 and (d) $\frac{1}{2}:\frac{1}{4}:\frac{1}{4}$.

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
