# Peer review of "New sensitivity of current LHC measurements to vector-like quarks"

_SciPost Physics, doi:SciPost Phys. 9, 069 (2020)_

## Round 1 · Referee Report · Anonymous (Referee 1) · 2020-6-23

Strengths

1- Broad results, complementary to standard analyses 2- The methodology used can be easily exported to other experimental analyses and have significant positive implications.

Weaknesses

1- No significant weaknesses

Report

The article uses the impressive Rivet+Contur framework to study a vast array of LHC measurements and searches on models with new vector-like quarks. Searches (or measurements) that are not specifically designed for these models, and quite often control regions in such searches, are shown to have a significant reach. They are often complementary to the more specific searches and therefore an excellent test of these models. Furthermore, in some cases they reach essentially make the specific searches redundant. The automated process involved allows one (and in particular big experimental collaborations) to explore the improvement that a specific search has on particular models over the implications of generic measurements or searches. In my view, this is a very strong work that deserves to be published in SciPost Physics and will likely have a significant impact on the field. I have only some minor, almost aesthetic, comments on the current draft:

1- The fermionic arrow corresponding to $Q_j$ in Fig. 1 (e) should point in the opposite direction. 2- The sentence in the middle of page 5 "... production of any flavour of VLQ in association with a V becomes suppressed with respect to pair-production, ..." can lead to misunderstanding from the reader as it does not take into account the phase space suppression in pair production versus single production, which I know the authors are aware of because they comment on this in section 5. Maybe a short comment here would be useful. 3- I'm not sure I fully understand the comment about the suppression of the $T+q$ and $B+q$ channels mentioned at the top of page 6. The comment about the u and d Yukawa couplings would suggest a suppression of several orders of magnitude. Is that true? Is there no $Z$ contribution? Aren't the charged current channels also suppressed by $\xi$ factors?

Requested changes

1- I would suggest to fix Fig 1 (e) and maybe clarify my points 2 and 3 in the report.

---

## Round 1 · Referee Report · Anonymous (Referee 2) · 2020-7-27

Strengths

1- Highlighting a novel way to constrain a class of models, beyond dedicated searches 2- coverage of a broad class of new physics models 3- use of a new source of data, in the form of standard model measurements

Weaknesses

1- the configurations considered in the manuscript are not exactly "realistic" (see full report) 2- the results are not easily transferable to other "more realistic" cases, so it is difficult to evaluate the impact of the results in the paper

Report

The manuscript presents a new way to constrain new physics by use of precise standard model measurements. It focuses specifically on vector-like quarks, which are widely present in many new physics models and are already extensively searched for at the LHC. By use of CONTUR and RIVET data, the authors show that relevant constraints can be obtained, which in some case are competitive with, and complementary to, the dedicated searches.

The manuscript is mostly well written and clear (even though some statements need to be amended, see below), and contains relevant results. However, before recommending the manuscript for publication, the authors need to address some important questions.

My main concern is that the configurations considered by the authors are not realistic, in the sense that they do not occur naturally in any concrete model. Firstly, as a benchmark case with all three branching ratios, the authors consider W:Z:H=1/3:1/3:1/3, however this case do not occur. The couplings of the VLQ with the Z, W and H are mainly governed by the weak isospin of the electroweak multiplet it belongs to. It turns out that for singlets and triplets we have W:Z:H=1/2:1/4:1/4, while for doublets we have W:Z:H=0:1/2:1/2. This is explained in Ref.[2] of the manuscript, and is valid for all mixing patterns and for any number of VLQ multiplets.

Finally, in some plots the authors consider a scenario where all types of VLQs are present and degenerate: X, T, B and Y. While I agree that degenerate VLQs are possible and likely to exist, the scenario chosen by the authors does not occur. In fact, the number and type of VLQs depend on the electroweak multiplet that can couple to the SM via the Higgs boson, and this has been classified first in hep-ph/0007316: one has singlets, T or B, doublets (X,T) or (T,B) or (B,Y), and finally triplets (X, T, B) or (T, B, Y). A quadruplet (X, T, B, Y) can only couple to the SM quarks via another VLQ multiplet (see 1502.00370) or higher order operators (see 1908.08964). In both cases, the degeneracy is likely to be broken and additional decays present.

One issue about the results shown in the various plots, which are based on one or more of the "unrealistic" cases, is that they cannot be transported to more realistic cases, thus their impact on the field is very limited. People interested would need to run CONTUR themselves, instead of reading this paper.

I therefore suggest the authors to revise the benchmark cases they consider, and target more realistic ones.

Requested changes

  • As discussed in the main report, the assumption W:Z:H=1/3:1/3:1/3 is not realistic. Typically, one has 1/2:1/4:1/4 for singlets and triplet, and 0:1/2:1/2 for doublets (in the large MQ limit). I would suggest the authors to consider one (or both) of this more realistic cases instead of 1/3:1/3:1/3. This change will make their results more relevant and reusable.

  • In the second paragraph on page 3, a discussion of he coupling structure is presented. There is the suggestion that the decay into H may be enhanced by the MQ dependence of the coupling. This is incorrect: the presence of the MQ/gv term distinguishes a scalar from a vector. In other words, if one has a vector coupling ~g and a scalar coupling ~ MQ/gv, than the BRs into Vq and Hq are of the same order. The MQ dependence is included in Ref[2] exactly to make sure that the BRs are equal to combination of the zeta and xi parameters. This paragraph need to be amended.

  • Are the cross-sections shown in Figs 1,2,3 at LO or NLO in the QCD coupling? The authors should be aware that NLO results are available for both pair and single production at NLO. Incidentally, the labels in the figures are too small to be readable.

  • In section 3, a comparison between the CONTUR result and ATLAS dedicated searches is presented. Some crucial information is lacking to fully appreciate the significance of this results. What are the integrated luminosities of the ATLAS searches versus those in CONTUR? Also, the authors should comment on the CMS direct searches: do they cover the area excluded by CONTUR?

  • In section 4, it is not clear at all what scenario is being considered. Are the VLQ only coupling to third generation? If yes, the axes labels in Fig.6(a) are misleading. Also, a scenario with degenerate X, T, B and Y is considered. This is however not realistic at all. I would suggest the authors to choose a more realistic case, by choosing a consistent electroweak multiple (see main report for details). I would also like to point out 1405.0737 where a strategy to extend the bounds to multiple VLQs is presented, and a comparison with SUSY searches also presented. This goes a little bit along the same lines of this paper.

  • Again in Section 4, there is one missing point: the direct bounds will also be enhanced by the presence of multiple degenerate VLQs (see for instance 1405.0737). The authors need to comment on this, else the reader may get the incorrect impression that CONTUR bounds overrun the direct searches.

  • Similar comments about the choice of realistic configurations apply to the results in Section 5. I'd suggest the authors to choose an EW multiplet, if they want to insist on adding more than one VLQ (degenerate). About the branching ratios, it is very useful to show the exclusive cases 1:0:0, 0:1:0 and 0:0:1. However, the democratic one is not very useful. I would suggest the authors to consider the more realistic 1/2:1/4:1/4 and 0:1/2:1/2 (either one of them, or both).

  • About the results presented in Section 5: are all the possible production modes in Fig.1 included in the analysis? How about (d), which contains model-dependent gauge couplings?

  • In the Introduction, the names of the VLQs are introduced: I would suggest to define here already their charge.

  • On page 4, the authors point out the relevance of electroweak pair production in case of couplings to first generation. This was first pointed out in 0911.4630 in a specific model. The authors should also comment on the fact that the process in diagram (d) contains a gauge coupling (of Z and W) which is model dependent, i.e. it depends on the electroweak quantum numbers of the multiplet the VLQ belongs to.

  • In diagram (e), the arrow on the bottom Q should be reversed.

---

## Round 2 · Referee Report · Anonymous (Referee 1) · 2020-9-15

Strengths

1- Broad results, complementary to standard analyses 2- The methodology used can be easily exported to other experimental analyses and have significant positive implications.

Weaknesses

1- No significant weaknesses.

Report

I think the improvements introduced in the second version of the article make it suitable for publication in SciPost.

Incidentally, I'm a bit confused by the choice of 0:1/2:1/2 branching fractions for the doublet case. I understand it was motivated by the request by the reviewer but this BRs only make sense if one is considering one of the two quarks in the doublet. If both quarks in the doublet are included, then we recover the 1/2:1/4:1/4 ratios when adding both quarks (as dictated by the equivalence theorem). For instance in the (X, T) case Gamma(X->Wt)=2Gamma(T->Z t)=2Gamma(T->H t) in the asymptotic limit. Probably this is what the authors do (they explicitly say that the X->Wt BR is always 1) but it is not entirely obvious from the text in the appendix. In any case, I think this is a minor issue that I don't feel needs any change.

---

## Round 2 · Referee Report · Anonymous (Referee 2) · 2020-10-7

Strengths

1- Highlighting a novel way to constrain a class of models, beyond dedicated searches 2- coverage of a broad class of new physics models 3- use of a new source of data, in the form of standard model measurements

Weaknesses

None

Report

The authors have replied to all issues raised in the previous reports, so I think that the manuscript can be published in the current form. I particularly appreciate that the authors provided many new plots in the Appendices, as they provide very useful information for establishing bounds on this class of models.

Requested changes

None

---

## Round 2 · Author Response

We thank the editors and referees for the positive comments and helpful suggestions, which we have implemented, and which we believe have helped us to improve the manuscript. In particular, the comments regarding the most well-motivated VLQ multiplets and W/Z/H admixtures have led to some important improvements to the paper. We have decided to add many additional results in the appendix which cover the scenarios mentioned (namely, considering singlets, doublets and triplets, and changing to a WZH=011 or WZH=211 scenario as required, instead of an unrealistic WHZ=111 case). We have also changed the plots in the main body to represent the WZH=211 case instead of the democratic one. We have kept the plots with four VLQs in the main body as the maximal case study, but the comparison with other results shows that the overall message is unchanged, and the more realistic cases are now available in the appendix, which we hope will satisfy the reviewers.

---

## Round 2 · List of Changes

We have introduced some other changes to the manuscript as requested, with details provided below.

In the introduction, we have specified the charges of the VLQs, as suggested. The reviewers asked whether the signal predictions were LO or NLO. We confirm they are all LO, and mention this in the text and relevant figure captions (where we also increased the label size as requested). We are aware that NLO predictions exist and have now mentioned them, but using those is beyond the scope of this work.

For Section 2, in the second paragraph , we agree with the reviewers that the statement was confusing, and we have opted to delete the statement, which in the end was not really needed. We now note that the effect of first-generation quark couplings on EW pair-production has been remarked upon in the previous papers. We have remarked upon the W/Z -> QQ coupling as suggested. The arrow on the diagram in fig 1 e) has been fixed. Also in this section, as requested, we have clarified/reworded the statements on: phase space suppression in pair production versus single production; and the suppression T+q/B+q channels.

In Section 3, when comparing to LHC results, we have noted the integrated luminosities of the relevant analyses, and made a statement and added references for the CMS coverage.

Regarding Section 4/Fig 6, we have clarified that this relates to third-generation couplings only. We take on board the comment about the BTXY-multiplet being unrealistic, and now provide results for the various singlets, doublets and triplets which are allowed (in addition to the unrealistic BTXY case as a benchmark). As suggested, we have also commented that these bounds cannot be directly compared to searches without accounting for enhancements due to degenerate VLQs.

For section 5, we have kept in the main body the BTXY results as a useful benchmark, but noted that the more realistic multiplet results can be found in the appendix. We have also added Section 5.4 which describes the main differences between the BTXY and multiplet results. We have also replaced the WHZ=111 with a more appropriate admixture in each case. We have clarified as requested which diagrams from Fig 1 are included in the study.

In addition, we have added additional references where appropriate.

---

## Editorial Decision

published